# Obesity in Adults: Position Statement of Polish Association for the Study on Obesity, Polish Association of Endocrinology, Polish Association of Cardiodiabetology, Polish Psychiatric Association, Section of Metabolic and Bariatric Surgery of the Association of Polish Surgeons, and the College of Family Physicians in Poland

**DOI:** 10.3390/nu15071641

**Published:** 2023-03-28

**Authors:** Magdalena Olszanecka-Glinianowicz, Artur Mazur, Jerzy Chudek, Beata Kos-Kudła, Leszek Markuszewski, Dominika Dudek, Piotr Major, Piotr Małczak, Wiesław Tarnowski, Paweł Jaworski, Elżbieta Tomiak

**Affiliations:** 1Health Promotion and Obesity Management Unit, Department of Pathophysiology, Medical Faculty in Katowice, Medical University of Silesia, 40-055 Katowice, Poland; 2Institute of Medical Sciences, Medical College of Rzeszow University, University of Rzeszów, 35-601 Rzeszow, Poland; drmazur@poczta.onet.pl; 3Department of Internal Diseases and Oncological Chemotherapy, Medical Faculty in Katowice, Medical University of Silesia, 40-027 Katowice, Poland; 4Department of Endocrinology and Neuroendocrine Tumors, Department of Pathophysiology and Endocrinology, Medical University of Silesia, 40-055 Katowice, Poland; 5Faculty of Medical Sciences and Health Sciences, University of Humanities and Technology in Radom, 26-600 Radom, Poland; 6Department of Psychiatry, Jagiellonian University Medical College, 31-008 Cracow, Poland; 72nd Department of General Surgery, Jagiellonian University Medical College, 31-008 Cracow, Poland; 8Department of General, Oncological and Bariatric Surgery, Centre of Postgraduate Medical Education, Orłowski Hospital, 00-416 Warsaw, Poland; 9The College of Family Physicians in Poland, 00-209 Warsaw, Poland

**Keywords:** obesity, adults, joint statement, primary care

## Abstract

Obesity in adults and its complications are among the most important problems of public health. The search was conducted by using PubMed/MEDLINE, Cochrane Library, Science Direct, MEDLINE, and EBSCO databases from January 2010 to December 2022 for English language meta-analyses, systematic reviews, randomized clinical trials, and observational studies from all over the world. Six main topics were defined in the joint consensus statement of the Polish Association for the Study on Obesity, the Polish Association of Endocrinology, the Polish Association of Cardio-diabetology, the Polish Psychiatric Association, the Section of Metabolic and Bariatric Surgery of the Society of Polish Surgeons, and the College of Family Physicians in Poland: (1) the definition, causes and diagnosis of obesity; (2) treatment of obesity; (3) treatment of main complications of obesity; (4) bariatric surgery and its limitations; (5) the role of primary care in diagnostics and treatment of obesity and barriers; and (6) recommendations for general practitioners, regional authorities and the Ministry of Health. This statement outlines the role of an individual and the adequate approach to the treatment of obesity: overcoming obstacles in the treatment of obesity by primary health care. The approach to the treatment of obesity in patients with its most common complications is also discussed. Attention was drawn to the importance of interdisciplinary cooperation and considering the needs of patients in increasing the long-term effectiveness of obesity management.

## 1. Introduction

The World Health Organization (WHO) recognized obesity as a disease in the last century. It was included in the International Classification of Diseases (ICD-10) under number E66 [1]. In recent years, the incidence of overweight and obesity has been systematically increasing, and the time of the COVID-19 pandemic caused a deterioration in the mental health of societies. Numerous studies indicate that many people began to deal with negative emotions with food during the pandemic, which may result in a further increase in the incidence of obesity. Despite this, obesity is one of the diseases rarely diagnosed and even less frequently treated. As international research shows, one of the reasons is the lack of knowledge of doctors about the diagnosis and treatment of obesity. Patients with obesity experience inequalities in health and limitations in self-determination not only because of the underlying disease but also when they develop other chronic diseases due to the lack of equipment, negative attitudes of medical staff caused by stereotypical thinking, lack of knowledge in the field of obesity treatment, and inability to refer the patient to the obesity treatment center [2]. Numerous global and European guidelines on the diagnosis and treatment of obesity signed by various societies were published. However, cultural and healthcare organizational differences require adaptation of recommendations at national levels. This is the first joint position statement of the Polish Association for the Study on Obesity, the Polish Association of Endocrinology, the Polish Association of Cardiodiabetology, the Polish Psychiatric Association, the Section of Metabolic and Bariatric Surgery of the Society of Polish Surgeons, and the College of Family Physicians in Poland. The expert panel’s goal was to develop comprehensive, evidence-based guidelines addressing the prevention, diagnosis, and treatment of obesity and its complications in adults. The aim of the recommendation was to assist health care providers—family doctors, nurses, physiotherapists, registered dietitians, and psychologists in diagnostics and effective treatment of obesity in adults. 

## 2. Methods

This search was conducted by using PubMed/MEDLINE, Cochrane Library, Science Direct, MEDLINE, and EBSCO databases from January 2010 to December 2022 for English language meta-analyses, systematic reviews, randomized clinical trials, and observational studies from all over the world. The websites of scientific organizations, such as WHO and EASO, were also searched. Six main topics, restricted to adults, were defined: (1) definition, causes, and diagnosis of obesity; (2) treatment of obesity; (3) treatment of main complications of obesity; (4) bariatric surgery and its limitations; (5) the role of primary care in diagnostics and treatment of obesity and barriers; and (6) recommendations for general practitioners, regional authorities and the Ministry of Health.

## 3. Obesity—Definition, Causes, and Consequences

### 3.1. Obesity—Definition

The WHO defines obesity as excessive or abnormal fat accumulation causing deterioration of health. In turn, overweight is named pre-obesity when the excess of fat does not yet meet the criteria for diagnosing obesity. Obesity is a chronic disease without a tendency to spontaneously resolve and with a tendency to relapse. The etiology of obesity is complex, and various causative factors are a reason for increased food consumption not balanced by physical activity, which results in a long-lasting positive energy balance and storage of the excess energy in adipose tissue [3]. 

#### 3.1.1. Diagnostic Tools and Data Interpretation

Despite numerous reservations about the sensitivity of body mass index (BMI) in diagnosing obesity and factors that may affect the false positive or negative result, this simple indicator remains the main criterion for diagnosing overweight and obesity.

We currently have two BMI cut-off points for diagnosing obesity in adults, proposed by the WHO in 1998 [3] and by the American Association of Clinical Endocrinologists and the American College of Endocrinology in 2016 [4]. The comparison of these cut-off points is presented in Table 1. We recommend using the cut-off points proposed by the American Association of Clinical Endocrinologists and the American College of Endocrinology because the presence of complications indicates clinically overt obesity, not pre-obesity. 

#### 3.1.2. Determination of the Severity of the Disease According to the Judgment of the Clinician

It should also be noted that visceral obesity (abnormal intraabdominal fat accumulation related to a higher risk of obesity complications) must be diagnosed based on waist circumference measurement in subjects with a BMI from 18.5 kg/m^2^ to 35.0 kg/m^2^. Visceral obesity should be diagnosed in adults according to the International Diabetes Federation (IDF) cut-off points of waist circumference in women > 80 cm and in men > 94 cm (referring to Caucasians). Waist circumference should be measured at the level of the midpoint between the bottom edge of the lowest rib and the upper edge of the iliac crest [5]. 

If possible, body composition should be measured using the bioimpedance method. The percentage of fat mass > 25% in men and >35% in women allows for the diagnosis of obesity in adults, while values 20.0–24.9% in men and 30.0–34.9% in women allow for the diagnosis of overweight [3]. 

### 3.2. Causes of the Development of Obesity

As was mentioned above, excessive fat accumulation is the result of long-lasting positive energy balance. However, the primary causes leading to a positive energy balance may vary from patient to patient. The primary cause of excessive food intake diagnosis is necessary for proper therapeutic decision-making and the most effective treatment. 

#### 3.2.1. Environmental Factors

The civilization development that has taken place in recent decades is conducive to the creation of a positive energy balance. The widespread use of motorization and automation in everyday life and professional work has reduced energy demand related to decreased physical activity. At the same time, the structure of consumed food unfavorably changed towards highly processed and low in dietary fiber. The manufacturer’s focus on lowering prices and mass production has significantly reduced the quality of food products and, at the same time, their energy density increased. In the Polish population, there is a growing trend in the consumption of fat and sugar. In the analysis of energy consumption in 167 countries conducted in 2018, Poles were ranked 10th [6]. In addition, Polish research showed low fiber intake in the middle-aged Polish population (16 g per day, referring to the lowest recommended 25 g per day) [7]. 

In recent years, due to the observed social, cultural, economic, and political changes, a concept of an obesogenic environment has emerged. That is, the living environment of the individual promotes and strengthens the behavior of the individual leading to a positive energy balance [8]. In this context, attention should also be paid to factors shaping consumer behavior, such as food advertising aimed at children and parents. These advertisements often use a subliminal message indicating the health benefits of certain products or the promotion of sweets by sports celebrities. The dynamic of expenditures on advertising by fast food restaurants and companies producing sweets in Poland has an upward trend [8]. Numerous studies have also shown the influence of the work environment on eating behavior. These factors include lack of conditions for regular meals, reduced physical activity, long working hours resulting in a heavy evening or night meal, shift work, disturbed sleep patterns or work-related stress and job dissatisfaction, and difficulties experienced in the workplace, including discrimination [8]. 

These causes should be assessed on the basis of a carefully collected medical history. 

#### 3.2.2. Genetic Factors

Mutations within a single gene or chromosomal changes involving several genes may be the cause of congenital disease syndromes, most often multi-symptom, in which severe obesity develops already in childhood. Almost 100 such syndromes have been described, although some of them do not yet have a name and are not genetically well characterized, and some of them can be found in the literature under several terms. 

At least 7% of non-syndromic early-onset severe obesity (NESO) has been shown to be the result of a single gene mutation. Most of these mutations concern genes encoding enzyme or receptor proteins involved in the regulation of the leptin–melanocortin pathway in the hypothalamus (e.g., *LEP*, *LEPR*, *PC1/PC3/PCSK1*, *POMC*, and *MC4R* genes) or having a significant impact on the development of this part of the brain (*SIM1*, *BDNF*, and *NTRK2* genes). 

Obesity, conditioned by polymorphisms of many genes, is the most common form of the disease. From a population point of view, significant mutations of single genes or chromosomal aberrations are only a margin of the problem. There is no single ‘obesity gene’ responsible for the development of this condition. The research on the so-called ‘candidate genes’ related to the regulation of hunger, satiety, development of adipose tissue, energy expenditure, or metabolic changes allowed us to isolate several hundred genes whose specific single nucleotide polymorphisms (SNPs) only favor the development of obesity [8]. 

In the vast majority of cases of familial obesity, the reproduction of unfavorable eating habits and patterns of spending free time play a greater role than genetic factors. In addition, family dysfunction (too little or too much parental care, inability to show affection, and excessive parental demands) is an important risk factor for the development of eating disorders [9]. 

The diagnosis of monogenic obesity should be considered in patients with a history of third-degree obesity, which begins in early childhood, especially if the implementation of all adequate treatment methods does not achieve a therapeutic effect.

#### 3.2.3. Emotional Eating and Eating Disturbances (Binge Eating Syndrome and Night Eating Syndrome)

In recent years, numerous studies have focused on the psychological basis of the development of obesity. The main strands of these studies concern personality traits and the dysfunction of the reward system [10,11,12,13]. Not recognizing these disorders and not incorporating appropriate therapeutic methods may be the main cause of obesity treatment failures.

The personality traits related to the risk of the development of emotional eating and eating disturbances included impulsivity (tendency to act rapidly without consideration of consequences), disinhibition, neuroticism, extraversion, sensation seeking, inattention, insufficiency inhibitory control, and the lack of cognitive flexibility [10]. 

The biological aspect of food intake regulation (hunger and satiety) is associated with the response of neurotransmitters in the hypothalamus to hormonal signals from the digestive tract and adipose tissue. However, the second place affecting food intake and eating behavior is the reward system (the amygdala/hippocampus, insula, orbitofrontal cortex [OFC], and striatum). The dysfunction of the reward system, especially decreased dopamine secretion, is associated with feeling appetite, also named food craving (the need to eat for pleasure, not for hunger). Emotions play a significant role in triggering processes of motivation to seek reward, learning, and consolidation of eating behavior arise, while the cognitive control of eating behaviors is localized in the prefrontal cortex [14,15]. 

##### Emotional Eating (EE)

Emotional eating, formerly called stress eating is a noneffective strategy for dealing with emotions with food. Emotions cause stress in the body and activation of the hypothalamic-pituitary-adrenal axis. In turn, cortisol inhibits dopamine release in the reward system and slows down the inhibitory-control pathway [16]. 

The COVID-19 pandemic worsened human mental health. Numerous studies have shown that many people cope with negative emotions with food [17]. Thus, this cause of the development of obesity should be included in the diagnosis work-up. It should be noted that over time EE may worsen, and binge eating disorder may develop. 

All patients with obesity should be assessed for EE. In everyday clinical practice, the screening tool presented in Table 2 should be used [18].

##### Binge Eating Disorder (BED)

In accordance with the Diagnostic and Statistical Manual of Mental Disorders (DSM), the fifth edition, BED should be diagnosed if consuming unusually large amounts of food in a short time with a loss of control that occurs at least once per week for 3 months. In addition, at least three of the following must be present: consuming food more rapidly than normal, eating until uncomfortably full, consuming large amounts of food without the feeling of hunger, eating alone to avoid shame or feeling disgusted with oneself, and depression, or guilt after gluttony without any regular compensatory behavior [19]. 

Of note, BED may be a primary cause of the development of obesity and also may secondarily develop in people suffering from obesity as a result of using numerous short-term diets [20]. The extreme form of BED is food addiction. Symptoms of food addiction include a compulsion to eat food, lack of control over food intake, physiological withdrawal symptoms, development of tolerance (i.e., the need to eat more and more food), neglecting other activities that may give pleasure, denying that there is a problem with eating control, and continuing behaviors related to food intake despite knowing that they are harmful.

##### Night Eating Syndrome (NES)

NES is diagnosed in subjects with recurrent episodes of excessive food consumption after dinner or eating after awakening from sleep and at the least three of the following: morning anorexia, a strong urge to eat between dinner and sleep and/or during the night, sleep onset and/or maintenance insomnia, frequently depressed mood or mood worsening in the evening, and a belief that one cannot go back to sleep without eating [21]. 

All patients with obesity should be assessed for BED and NES (Table 3).

Both BED and NES often coexist with depression and anxiety. Depression and anxiety should be diagnosed in all patients with obesity based on Hospital Anxiety and Depression Scale (HADS). 

#### 3.2.4. Obesity Associated with Hormonal Disturbances

Obesity can develop in the course of some endocrinopathies, including:Cushing’s syndrome, ACTH dependent (Cushing’s disease), and ACHT independent;Hypothyroidism in the course of primary or secondary thyroid dysfunction;Pituitary dysfunction in the form of multihormonal hypofunction of this gland, including growth hormone deficiency;Damage to the hypothalamus with the impaired secretion of hypothalamic neurohormones.

The guidelines of the European Society of Endocrinology (ESE) from 2020 contain the current recommendations of the year regarding their diagnosis [22].

##### Cushing’s Syndrome

The prevalence of hypercortisolism in people with obesity is estimated at 0.9%.

Routine testing for hypercortisolemia is not recommended except when suspected on clinical examination (blue-red skin stretch marks, bruising, or proximal muscle weakness) and resistant hypertension. Conducting laboratory hormonal tests is not recommended in patients with iatrogenic Cushing’s syndrome, especially those undergoing chronic glucocorticoid therapy. In patients with obesity, in whom bariatric surgery is planned, tests to exclude hypercortisolemia should be considered [22,23].

Diagnostic tests:-Inhibition test with 1 mg dexamethasone;-Assessment of free cortisol concentration in a 24 h urine collection or late evening cortisol concentration in saliva;-If there is confirmed endogenous hypercortisolism, then measure ACTH levels and plan imaging tests [22,23,24].

##### Hypothyroidism

Overt hypothyroidism occurs in 14% of patients with obesity, subclinical hypothyroidism in another 14.6% of patients with obesity, and their frequency is significantly higher than in the general population (in Europe, overt hypothyroidism occurs with a frequency of 0.2–5.3%, and subclinical hypothyroidism is more often of 4–10%), the incidence of undiagnosed hypothyroidism is estimated at about 5%. The assessment of thyroid function is recommended in all patients with obesity [22,25]. 

Diagnostic tests:-Serum TSH levels as part of tests performed in all people with obesity, regardless of the presence of symptoms suggesting thyroid dysfunction;-Free thyroxine (FT_4_) and anti-thyroid peroxidase (anti-TPO) antibodies are recommended to be measured if elevated TSH is found.

The reference ranges for TSH and FT_4_ in patients with obesity are the same as in the general adult population [22,23]. 

An ultrasound examination of the thyroid gland is recommended for a full assessment of the thyroid gland, although the ESE guidelines do not require routine thyroid ultrasound examination in obese patients if no abnormalities are found in the physical examination of the thyroid gland.

##### Pituitary Dysfunction in the Form of Multihormonal Hypofunction of the Pituitary Gland, Including Growth Hormone (GH) Deficiency, and Rare Damage to the Hypothalamus with Impaired Secretion of Hypothalamic Neurohormones

They occur most often after surgery or radiotherapy in the area of the hypothalamus and pituitary gland and may be caused by compression (tumors, craniopharyngioma, or metastases), ischemia, trauma, sarcoidosis, storage diseases (hemochromatosis and histiocytosis), autoimmunity (lymphocytic hypophysis) and infectious factors.

Diagnostic tests:-Serum GH, FSH and LH, TSH, ACTH, and PRL levels;-Serum insulin-like growth factor type 1 (IGF-1), estradiol, testosterone, cortisol, fT_3_, and fT_4_ levels;-Stimulating tests (with insulin, with arginine, with GH-RH, with LH-RH, and with CRH) [22,23].

#### 3.2.5. Medication-Related Obesity

##### Glucocorticoids

Weight gain occurs in approximately 70% of patients treated with glucocorticoids, of which 20% exceed 10 kg. The effect of glucocorticoids on food intake is complex and includes both changes in the secretion of neurotransmitters responsible for the regulation of satiety and hunger in the hypothalamic nuclei, as well as neurotransmitters responsible for the hedonic aspect of food intake in the reward system [8].

##### Hypoglycemic Drugs

Hypoglycemic drugs promoting weight gain include insulin, insulin analogs, sulfonylureas, and thiazolidinediones. 

Weight gain during insulin and insulin-analogs use is dose-dependent, related to stimulation of food intake, episodes of hypoglycemia, and fluctuating glucose concentrations. Many patients eat not only when symptoms of hypoglycemia appear but also because they are afraid of their occurrence.

Sulfonylureas stimulate the secretion of endogenous insulin, which may result in hypoglycemia and significant fluctuations in glucose levels, and, consequently, in increased food intake. 

Weight gain also occurs with thiazolidinediones and is dose-proportional. The weight gain effects of these drugs include fluid retention, increased storage of triglycerides in adipocytes, and enhanced adipogenesis. Interestingly, the accumulation of adipose tissue primarily occurs in the visceral deposit [26].

##### Antihypertensive Drugs

It has been known for many years that the use of beta-adrenergic antagonists (except carvedilol and nebivolol) causes weight gain in some people, and it is associated with genetic variants of the beta-adrenergic receptors. These drugs decrease energy expenditure—the basal metabolic rate by 4–9% and postprandial thermogenesis by 25%. They also inhibit the activity of hormone-sensitive lipase and, in consequence, lipolysis. In addition, one of the side effects may be weakness and fatigue and, in consequence, decreased physical activity [26].

##### Psychotropic Medication

Neuroleptics

Up to 80% of patients on atypical neuroleptics gain 20% or more of their normal weight.

These drugs increase food intake by affecting the reward and punishment systems and increasing appetite. This is the result of their antagonistic effect on dopaminergic type 2 (D2) and serotonin type 2A (5-HT2A) receptors. They can also significantly affect histamine (H1) receptors and, to a lesser extent, α_1_-adrenergic and serotonergic type 2C (5-HT2C) receptors. The fact that there are differences in the amount of weight gain when using different drugs of this class is related to their potency in blocking the activity of particular receptors. The risk of weight gain during neuroleptics use is presented in Table 4 [27,28].

Antidepressants

It should be noted that in a quarter of patients using antidepressants, weight gain is observed. The risk factors of weight gain associated with antidepressant use include the type of medication, duration of pharmacotherapy, female sex, and overweight or obesity before initiation of the treatment. The risk of weight gain related to antidepressant use is presented in Table 5 [29]. 

Weight gain may also be experienced by patients with bipolar disorder, treated with lithium or valproate. 

Antiepileptic drugs

Weight gain has been observed in 71% of those treated with valproic acid and 43% of those treated with carbamazepine. Weight gain occurred less frequently when treated with pregabalin and gabapentin. There was no change in body weight during treatment with lamotrigine, levetiracetam, and phenytoin. On the other hand, felbamate, topiramate, and zonisamide cause weight reduction through an unknown mechanism [31].

### 3.3. Consequences and Complications of Obesity

Obesity is a chronic disease that can lead to disability. Musculoskeletal, cardiovascular, and mental diseases are the three most common reasons for people with obesity to enter a disability pension. Moreover, people with obesity are more likely to lose their jobs, retire more often, take sick leave more often, are less productive at work, and are more likely to be injured in the workplace. In addition, often, people with childhood obesity develop physical disabilities at a young age and do not enter the labor market at all. It was also shown that being obese at the age of 18 increased the risk of taking disability benefits by 35%. It has also been observed that an increase in BMI by 1 kg/m^2^ increases the risk of physical disability by 5%. Factors that increase the risk of developing disability in patients with obesity are anxiety and depressive disorders [32,33,34]. 

A systematic review of studies conducted in European countries showed that patients with obesity took about 10 days longer sick leave per year than those of normal weight. The risk of taking sick leave lasting from 2 weeks to 12 weeks was 34% higher among patients with obesity and longer than 3 months by 63% [35]. 

Being overweight or obese predisposes to the development of numerous dangerous complications, including metabolic, mechanical, and other. 

#### 3.3.1. The Metabolic Complications of Obesity

These complications developed as a result of excessive accumulation of visceral adipose tissue with local inflammation, adipokine secretion disturbances, and insulin resistance. The adipose tissue becomes inefficient in the field of energy storage and comes to ectopic accumulation of fat in the liver and skeletal muscle and the development of insulin resistance. Systemic inflammation, changes in adipokine secretion, insulin resistance, and hyperinsulinemia are the key links in the development of obesity complications in adults, such as: Nonalcoholic fatty liver disease (NAFLD), currently called metabolic-associated fatty liver disease (MAFLD);Pre-diabetes (impaired glycemia fasting [impaired fasting glucose (IFG)] and impaired glucose tolerance [impaired glucose tolerance (IGT)] and type 2 diabetes;Atherogenic dyslipidemia (decreased HDL-C, elevated TG, at frequent slight changes in TC and LDL-C concentrations);Cardiovascular diseases (hypertension, coronary artery disease, carotid atherosclerosis, and stroke);Obesity-induced glomerulopathy;Cancers (e.g., colon, breast, and endometrium);Hormonal disturbances that lead to infertility in women (functional hyperandrogenism and polycystic ovary syndrome [PCOS]) and men (hypogonadism).

##### Non-Alcoholic Fatty Liver Disease (NAFLD)/Metabolic-Associated Fatty Liver Disease (MAFLD) 

The diagnostic criteria of NAFLD are hepatic steatosis > 5% and exclusion of secondary causes of liver disease, including ‘significant’ alcohol usage. While the diagnostic criterion for MAFLD formulated in the year 2020 by an expert group utilizing a two-stage Delphi consensus is hepatic steatosis > 5% and metabolic risk divers, such as type 2 diabetes and overweight/obesity by ethnic-specific BMI classifications. In people with normal weight for the diagnosis of MAFLD, hepatic steatosis > 5% and two of seven risk factors are needed, including waist circumference > 102 cm in Caucasian men and >88 cm in Caucasian women; blood pressure > 130/85 mmHg or hypotensive therapy; plasma triglycerides > 150 mg/dL or specific drug treatment; plasma HDL cholesterol < 40 mg/dL for men and <50 mg/dL for women or specific drug treatment; prediabetes (fasting glucose levels 100–125 mg/dL or 2 h post-load glucose levels 140–199 mg/dL or HbA1c 5.7–6.4%); homeostasis model assessment of insulin resistance score > 2.5; and plasma *C*-reactive protein levels (high sensitivity CRP) > 2 mg/L [36]. 

In all patients with overweight or obesity, an ultrasound of the liver should be performed. In patients with normal weight, the risk factors should be assessed. 

MAFLD is a progressive process from steatosis by inflammation and fibrosis to cirrhosis or hepatocellular carcinoma. However, the leading causes of premature death among people with NAFLD are cardiovascular complications [37]. The noninvasive method of fibrosis assessment is the FIB4 test (age, activity of ALT and AST, and the number of platelets) [38]. 

In patients with MAFLD, the main method of treatment is the effective management of obesity. However, the weight reduction should be no more than 0.5 kg per week. Too fast weight loss induces the formation of lithogenic bile and may cause increased fatty liver [4]. There is no safe amount of alcohol for MAFLD [39]. If indicated, pharmacotherapy appropriate to the severity of carbohydrate and lipid metabolism disturbances caused by MAFLD should be used [40]. The use of ursodeoxycholic acid (UDCA) in a dose of 10–15 mg/kg/day should be considered [41].

##### Prediabetes and Type 2 Diabetes

Prediabetes includes impaired fasting glucose (IFG) related to fatty liver and its insulin resistance and impaired glucose tolerance (IGT) related to muscle fat and its insulin resistance. Disturbances associated with IFG result in an increase in hepatic glucose production and fasting hyperglycemia, but during activity, its plasma concentration gradually decreases as it is used as energy by muscles that remain insulin sensitive. In isolated IGT, on the other hand, muscle insulin resistance and a defect in the second phase of insulin secretion result in long-term hyperglycemia. In addition, the post-prandial release of glucagon-like peptide-1 is impaired, resulting in decreased insulin secretion [42]. 

It is estimated that about 70% of people with prediabetes will develop type 2 diabetes in the future if obesity is not effectively treated.

Prediabetes is an early stage in the development of type 2 diabetes. The progression of carbohydrate metabolism disorders towards diabetes is associated with impaired compensatory insulin secretion due to increased apoptosis of pancreatic islet β cells, which is facilitated by both impaired GLP-1 secretion and increased release of pro-inflammatory cytokines and leptin by visceral adipose tissue, as well as increased glucagon secretion and increased hepatic glucose synthesis as well as progressive changes in skeletal muscle metabolism [42]. 

All patients with overweight, obesity, or visceral obesity should be screened for IFG by testing fasting blood glucose at least 12 h after the last meal. Patients with fasting glucose levels 100–125 mg/dL should have an oral glucose tolerance test of 75 mg. Diabetes should be diagnosed based on the criteria of the Polish Diabetology Society [43].

The main method of treatment in patients with prediabetes and type 2 diabetes is the effective management of obesity. Metformin is recommended in patients with prediabetes as first-line therapy in most patients with type 2 diabetes. Other classes of hypoglycemic agents are useful in combination with metformin or when metformin is a contraindication or not tolerated. Their selection should base on the balance between the efficacy and side effect profile. All patients with type 2 diabetes and established or subclinical cardiovascular disease should be treated with the GLP-1 RA class or SGLT2i class [44]. 

##### Atherogenic Dyslipidemia

Atherogenic dyslipidemia is characterized by elevated serum triglyceride levels of at least 150 mg/dL (~1.7 mmol/L), elevated serum levels of very low-density lipoprotein (VLDL)-rich triglycerides, and decreased HDL cholesterol in men below 40 mg/dL (~1 mmol/L) and in women below 45 mg/dL (~1.2 mmol/L). Serum LDL may be normal or elevated, with an increased percentage of oxidized particles (oxLDL). These abnormalities are the results of fatty liver and increased triglycerides and VLDL production, as well as decreased HDL cholesterol synthesis [36].

Atherogenic dyslipidemia is associated with a residual risk of developing coronary heart disease in patients with serum LDL levels of 70 mg/dL or less, to a similar or greater extent than in the overall group [45].

The diagnosis of atherogenic dyslipidemia is based on the assessment of the lipid profile. Measurement of lipid profile should be performed in all people over 40 years and in all younger persons with factors of cardiovascular risk, including obesity and MAFLD [46]. 

The main method of treatment of atherogenic dyslipidemia is the management of obesity. In addition, statins in combination with fibrates or omega-3 fatty acids should be used [46]. 

##### Arterial Hypertension

Obesity is a major risk factor for developing arterial hypertension. The links to the pathogenesis of obesity-induced hypertension are complex, but each of them is based on excess visceral adipose tissue. These include inflammation and endocrine dysfunction of adipose tissue, insulin resistance, endothelial dysfunction, increased sympathetic nervous system activity, activation of the renin–angiotensin–aldosterone system, dysfunction of the natriuretic peptide system, and a rare development of obesity-related glomerulopathy. The effect of these changes is increased cardiac output, peripheral vasoconstriction, and impaired pressure natriuresis (water and sodium retention and increased blood volume) [47].

The diagnosis of arterial hypertension should not be based on a single blood pressure measurement taken during a single visit. Exceptions are rare situations in which blood pressure is significantly elevated (grade 3 arterial hypertension) or if there is clear evidence of complications of arterial hypertension (e.g., left ventricular hypertrophy, hypertensive retinopathy with effusions and petechiae, or kidney damage). In people with mean blood pressure values below 180/110 mmHg, arterial hypertension should be diagnosed on the basis of at least two blood pressure measurements taken during at least two separate visits. It should be noted that the basis for the diagnosis and treatment of arterial hypertension is still measurements made in the doctor’s office. However, arterial hypertension can also be diagnosed based on out-of-office measurements, i.e., ambulatory blood pressure monitoring (ABPM) and home measurements. In most patients, blood pressure should be measured using a standard arm cuff (width 12–13 cm, length 35 cm); if the patient’s arm circumference is >32 cm, a larger cuff should be used. At least 30 min before the measurement, the patient should refrain from consuming coffee, smoking cigarettes, and taking other stimulants. The measurement should be performed after at least five minutes of rest in a sitting position with the back supported in a quiet room with maintained thermal comfort. To determine the value of blood pressure, calculate the average of the last two measurements; at least three times, pressure measurements should be taken as a standard performed during the same visit at 1–2 min intervals. If blood pressure varies between measurements (>10 mmHg), additional measurements should be taken. At the initial assessment, all patients should undergo an orthostatic test, taking blood pressure measurements at 1 and 3 min after the change from sitting to standing position [48].

The therapeutic goals in patients with obesity under 65 years are blood pressure values 120–129/70–79 mmHg, in patients aged 65–80 years 130–139/70–79, and in over 80 years 130–150/70–79 [49]. 

An important part of arterial hypertension management is weight reduction. Combined pharmacotherapy is recommended in obese patients. Combinations of an angiotensin-converting enzyme inhibitor (ACE-I) or angiotensin receptor blocker (ARB) with a diuretic or calcium channel blocker (CCB) should be used as first-line drugs. In the second step, if the therapeutic goal is not achieved, triple therapy is preferred (a combination of ACE-I or sartan with a CCB and a diuretic, with separate therapy when there are indications for treatment with β-blockers). In the third step, a fourth drug should be added [49]. 

##### Obesity-Related Glomerulopathy (ORG)

In patients with obesity, there is an increase in renal blood flow and glomerular filtration, resulting in dilation of the afferent glomerular arterioles. The links of ORG pathogenesis are glomerular hyperfiltration, insulin resistance and hyperinsulinemia, hyperleptinemia, reduced anti-inflammatory effect of adiponectin, and chronic inflammation. Hyperleptinemia results in an increased secretion of fibroblast growth factor β (TGF-β), which stimulates the proliferation of endothelial and mesangial cells and the overproduction of extracellular matrix. Hyperinsulinemia not only stimulates myocyte proliferation in the media of the arteries but also promotes glomerulosclerosis by stimulating collagen synthesis. An additional factor involved in the pathogenesis of ORG is dyslipidemia. Characteristic symptoms of ORG are proteinuria and gradual impairment of renal excretion. ORG is a progressive nephropathy, and the rate of its progression depends on the occurrence of complications of obesity, such as arterial hypertension and type 2 diabetes [50,51]. ORG is a rarely diagnosed entity based on kidney biopsy in patients with high-range proteinuria.

##### The Main Hormonal Disturbances

Growth hormone (GH) deficiency

A decrease in GH and IGF-1 levels may be considered as obesity complication in patients without a pituitary disease. The routine determination of GH and IGF-1 in patients with obesity is not recommended [22,23]. 

Hypogonadism in Men

It occurs in 32.7% (up to 45%) of men with obesity.

In all men with obesity, an assessment of symptoms of hypogonadism (decreased libido, erectile dysfunction, infertility, muscle weakness, gynecomastia, gynoid type of fat distribution, and androgenic hair loss) should be conducted. Hormonal work-up in men without these symptoms is not recommended.

Diagnostic tests:-Serum concentrations of total and free testosterone, sex hormone-binding globulin (SHBG), FSH, LH, and PRL.

The reference ranges for serum testosterone levels in men with obesity are age-specific. Hypogonadism is diagnosed in men with serum testosterone levels ≤ 11 nmol/L (3.2 ng/mL) with the presence of symptoms [22,23,52].

Functional hyperandrogenism in women and polycystic ovary syndrome (PCOS)

It occurs in 9.1–29% of women with obesity. The diagnostic is recommended only in women with menstrual disturbances, chronic anovulation, infertility, or/and symptoms of androgenization (hirsutism, androgenetic alopecia, or acne). 

Diagnostic tests:-Serum concentrations of FSH, LH, PRL, estradiol, total testosterone, and SHBG (between 3–5 days of the menstrual cycle);-Concentrations of androstenedione, 17-hydroxyprogesterone, and progesterone (depending on individual indications).

Moreover, an ultrasound examination of the ovaries and determination of plasma glucose concentration is recommended [22,23,53]. 

#### 3.3.2. Diseases Caused by Mechanical Consequences of Excessive Accumulation Visceral Fat

##### Gastroesophageal Reflux (GERD)

Patients with obesity report numerous symptoms related to the function of the esophagus and stomach, including difficulties in swallowing, pain while eating or pain after eating, a feeling of fullness and retention in the stomach, heartburn, or regurgitation, which paradoxically does not translate into weight loss.

The factors contributing to the occurrence of these disorders in patients with obesity include increased intra-abdominal pressure and high support of the diaphragm as a result of the accumulation of visceral fat. Disorders are also favored by anatomical and functional abnormalities of the esophagus and stomach, causing abnormal esophageal motility and, thus, esophageal clearance, i.e., the ability to clean the esophagus from food residues or regurgitated contents, lowering the pressure of the lower esophageal sphincter, transient lower esophageal sphincter relaxation, or hiatal hernia [54]. 

All patients with overweight or obesity should be evaluated for symptoms of GERD, and in patients with these symptoms and if treatment fails to control symptoms, an endoscopy should be performed [4].

The treatment of GERD in patients with overweight or obesity includes at least 10% weight loss and the use of a proton pump inhibitor [4]. 

##### Obesity Hypoventilation Syndrome (OHS)

OHS is defined as the occurrence of symptoms of hypoventilation in patients with obesity when all other potential causes of hypoventilation have been excluded [55]. 

OHS is a result of hypoxemia observed during physiological sleep deepens and hypercapnia increases to pathological values that colloquially meet the definition of respiratory failure, i.e., a state in which the partial pressure of oxygen in arterial blood falls below 60 mmHg (PaO_2_ < 60 mmHg), or an increase in the PCO_2_ ≥ 45 mmHg. The primary pathomechanism of hypercapnia is hypoventilation. However, it should be emphasized that hypercapnia, in the case of pure, untreated OHS, is always accompanied by hypoxemia (type 2 respiratory failure). Compensation for respiratory acidosis is renal production of bicarbonates (HCO^3−^). The specificity of respiratory failure in the course of OHS is the fact that it begins to insidiously appear at night, especially during REM sleep, then appears and persists during NREM sleep, to finally become also consolidated during the day. In more advanced cases, respiratory disturbances occur around the clock and are no longer compensated by daily hyperventilation [55].

Clinical symptoms of OHS include impaired concentration, excessive daytime sleepiness, decreased exercise tolerance, and morning headaches [55].

All patients with obesity, especially II- and III-grade, should be evaluated for OHS. Obesity management is an essential element of therapy.

##### Sleep Apnea Syndrome (OSA)

OSA is the result of sleep-related decreased airflow and oxygenation. The increase in body weight significantly increased the risk of developing OSA. Neck circumferences above 40.6 cm in women and 43.2 cm in men are associated with an increased risk of OSA [56]. 

Symptoms include loud snoring, interruptions (apneic or hypopnea pauses) in breathing, and sleep-cycle fragmentation that, in turn, produce daytime fatigue, morning headache, lack of concentration, erectile dysfunction, and a general decrease in quality of life. In patients with these symptoms, polysomnography should be considered. Management of obesity is an essential part of therapy [57]. 

#### 3.3.3. Mechanical Damage Caused by Excessive Load

##### Osteoarthritis

Numerous studies indicate an indisputable relationship between obesity and the development of knee osteoarthritis. A correlation was found between the diagnosis of obesity and the development of various deformities of the knee, which is believed to be a mechanical factor in the development of obesity-dependent gonarthrosis [58].

Obese people adapt to their body weight by walking more slowly with their feet wider apart. They experience greater loads affecting the joints of the lower limbs, which predispose them to damage. Obesity is associated with structural disorders as well as impaired gait function, flattening of the arches of the foot, and excessive pronation in the ankle joint. When walking, there is an increase in the mobility of the rear foot, and this causes forefoot abduction to a greater extent than in a normal-weight person. Being overweight leads to increased pressure on the loaded joints. Postural instability leading to falls was found in people diagnosed with III-degree obesity [58].

Obesity is a significant risk factor for pain in the neck, shoulder, elbow, wrist, and hand. Obesity in professionally active individuals predisposes to the development of tendinitis in the upper limbs. Numerous studies indicate the risk of developing ulnar nerve groove syndrome or carpal tunnel syndrome in obese patients, especially those who perform repetitive activities during their professional activity. Obesity significantly increases the risk of rotator cuff tendinitis. Obesity is also a risk factor for greater trochanteric bursitis, a common cause of lateral hip pain in middle-aged and older adults [58].

Spinal pain syndromes caused by degenerative disc disease, stenosis of the spinal canal, and diseases of the intervertebral joints are very common problems in society, causing significant morbidity. This generates significant consequences for work efficiency and utilization of health services. The relationship between obesity and the described diseases is ambiguous. Some studies evaluating this issue find no evidence of a link between obesity and low back pain. However, compared with people with normal weight, obese patients more often suffer from radicular pain and present neurological symptoms [58].

Screening for symptoms and physical examination for osteoarthritis should be performed in all patients with overweight and obesity. Obesity management is an essential part of osteoarthritis treatment [4]. 

##### Chronic Venous Disease

Epidemiological studies have shown that obesity is the risk factor for varicose veins in both sexes. It has been suggested that the main mechanism of impairing venous function, particularly venous return, and possibly increasing the rate of reflux in patients with obesity is the high pressure in the abdomen [59]. 

#### 3.3.4. Other

##### Cholelithiasis

The risk of occurrence of cholesterol gallstone formation and symptomatic cholelithiasis increases significantly in patients with obesity and is augmented by weight loss, especially if it is fast. Approximately one-third of stones are symptomatic. The incidence of new gallstone formation is 10–12% after 8–16 weeks of application of a low-calorie diet and above 30% in the first 18 months after gastric bypass surgery. The higher risk of gallstone formation has also been observed in clinical trials that assessed the efficacy and safety of GLP-1 analogs. The additional risk factors for gallstone formation during weight loss include loss of more than 25% of the initial body weight, rate of weight loss above 1.5 kg per week, a very low-calorie diet containing little or no fat, and periods of absolute fasting. Cholelithiasis may be prevented by treatment with ursodeoxycholic acid 500–600 mg per day during the first 6 months of weight loss [4]. 

##### Stress Urinary Incontinence

Obesity is a major risk factor for urinary incontinence in women, and its frequency and severity increase with an increase in BMI values and duration of obesity. Screening for urinary incontinence should be performed in all women with overweight or obesity [4]. 

##### Asthma 

Asthma symptoms and severity are associated with increased proinflammatory cytokines and adipokines related to obesity. Numerous studies have shown improvement in forced vital capacity after an average 7.5% weight reduction in patients with obesity and asthma. 

Medical history, symptomatology, and spirometry should be considered in all patients with overweight or obesity with an increased risk of asthma and reactive airway disease [4]. 

##### Depression and Anxiety

Depression and bipolar disorder (BD) 

It has been shown that 30–50% of people seeking treatment for obesity had a history of depression or anxiety. The occurrence of depression symptoms in young women is an important risk factor for the development of obesity later in life. Higher BMI values than in the general population have already been observed in adolescents with depression. 

The association between depression and obesity seems bi-directional. The classic symptom of depression is loss of appetite and weight; however, when mood improves during the treatment, appetite and weight increase. Of note, in patients with obesity, more frequent atypical depression is observed. People with this type of depression deal with negative emotions with food. Food stimulates the release of dopamine in the reward system and temporarily improves mood. In this mechanism, depression may be the cause of the development of eating disorders. On the other hand, obesity may be a cause of the development of depression due to low self-esteem, discrimination, stigmatization, and social exclusion.

Therapy in patients with obesity, especially for bipolar disorder, is often less effective than in patients with normal body weight. Some studies have shown that bipolar disorder in patients with obesity is associated with a greater degree of disability, including impairment of memory, concentration, and attention, as well as a greater relapse rate and a more severe course of the disease [60]. 

The consequences of the coexistence of depression and obesity included worse patient–doctor cooperation, avoidance, and social withdrawal, decreased quality of life, greater severity of depression, greater risk of disability and job loss, suicidal thoughts, and attempts. In patients with obesity, anxiety disorders such as panic attacks and agoraphobia (fear and avoidance of being out in the open and in public places) is twice as common as in normal-weight people.

All patients with obesity should be screened for symptoms of depression and anxiety in the GP’s practice using the Hospital Anxiety and Depression Scale. Body weight and metabolic parameters should be monitored in all patients treated for psychiatric diseases. The family doctor should stay in touch with the psychiatrist and undertake joint actions aimed at the effective treatment of mental illnesses and limiting its consequences for physical health.

## 4. Obesity in Patients with Schizophrenia in a GP’s Practice

Obesity is diagnosed three times more often in patients with schizophrenia before the start of pharmacotherapy than in the general population. Treatment with neuroleptics is associated with a further increase in the risk of developing obesity. Moreover, schizophrenia per se is a cause of low physical activity, spending more time in bed, consumption of poor-quality food, and frequent smoking. The prevalence of components of metabolic syndrome among patients with schizophrenia is estimated at 37.0–63.0%, while in the general population, at 20.0–25.0%. Thus, at the start of the antipsychotic treatment, next to effectiveness and tolerability, is the impact of the drugs on food intake and, consequently, the development of obesity [61,62]. 

In the course of treatment, schizophrenia should be monitored: The presence of risk factors of or clinically overt cardiovascular disease (CVD) and/or diabetes mellitus, family history of CVD, smoking status, eating habits, and level of physical activity;Body weight and height with calculated BMI, waist circumference, and blood pressure (mean value of at least two measurements during a single visit);Fasting glucose, lipid profile, serum creatinine with estimated glomerular filtration rate (GFR), and uric acid level (Table 6) [46,63].

## 5. Treatment of Obesity

### 5.1. Therapeutic Goals

Establishing a therapeutic goal should meet the SMART business goal rule, i.e., specific, measurable, achievable, relevant, and timely. 

The overriding goal of obesity treatment is to slow down the progression of the disease, avoid relapses, and prevent the development of complications caused by excess fat in the body or reduce their severity, as well as overall improvement of the patient’s health and quality of life, and life extension. 

The overriding goal in patients without complications of obesity is to reduce the severity of the disease by one stage. While in patients with complications, this will be such a reduction in body weight that will contribute to a significant improvement in the control of these complications and will enable the reduction of doses and/or the number of drugs used, and in some less advanced cases will allow for the remission of complications and discontinuation of pharmacotherapy. 

Achieving such goals requires individual determination of the percentage reduction of body weight in relation to the initial value. The goal should always be set in such a way that the patient does not feel that it is so distant as to be almost unattainable. Therefore, it is worth setting 3–6 months stages, in which the goal is to reduce body weight by 5–10% of the initial one, followed by a 3–6 months period of maintaining the obtained results and, if necessary, another stage of 5–10% body weight reduction [66]. 

It is believed that different percentages of initial body weight reduction are required to improve individual complications of obesity:Approximately 10–40% in body weight reduction in patients diagnosed with non-alcoholic steatohepatitis in the course of MAFLD.At least 5% to 15% in body weight reduction in patients diagnosed with the following:
-Type 2 diabetes (lower HbA1c, reduce the number and/or doses of hypoglycemics drugs used, and remission of the disease, especially if it lasts a short time).-Dyslipidemia (decrease in blood triglycerides and non-HDL cholesterol, and increase in HDL cholesterol).-Arterial hypertension (reduction of systolic and diastolic pressure and reduction of the number and/or doses of antihypertensive drugs).-Polycystic ovary syndrome (return of ovulatory cycles and regular menstruation, reduction of hirsutism, improvement of insulin resistance, and reduction of androgen levels in the blood).At least 5% to 10% in body weight reduction is recommended in patients diagnosed with the following:
-Male hypogonadism (increased testosterone levels in the blood).-Stress urinary incontinence (reduced frequency of episodes of incontinence).At least 7–8% in body weight reduction is recommended in patients diagnosed with bronchial asthma (improvement in terms of forced expiratory volume in 1 s and reduction in the severity of symptoms).At least 7–10% in body weight reduction is recommended in patients diagnosed with obstructive sleep apnea.At least 10% in body weight reduction is recommended in patients with the following:
-Prediabetes (preventing the development of type 2 diabetes and improving glucose levels).-Improving female infertility (return of menstrual ovulation cycle, pregnancy, and the birth of a live newborn).-Osteoarthritis (reduction of pain and improvement of motor function);-Gastroesophageal reflux (reduced symptoms).At least 5% in body weight reduction is recommended in patients with the following:
-Steatosis stage in the course of MAFLD (reducing lipid accumulation in the liver and improving metabolic function) [4].

It is very important to set partial goals both in terms of the effect and the changes leading to them because the small-step method allows the patient to better adapt to changes and does not put pressure on them to achieve the effects [66]. 

To avoid patient disappointment and discouragement, one should explain to them that it is for their health, and slow is beneficial (approx. 1 kg/week in the first month) and approx. 0.5 kg/week. in the following months), but permanent weight loss. The main reason for losing weight is improving health, not the number of kilograms lost. Slow but systematic weight loss as a result of the use of a balanced diet and increased physical activity lowers blood pressure, serum glucose, and lipid levels, improves the quality of life, and in many people with diseases accompanying, allows one to reduce the number of drugs used.

Too fast, significant weight loss causes significant loss of lean mass and increases the risk of developing gallstones and fatty liver, and occurrences of the ‘yo-yo’ effect. 

### 5.2. Rule of the Five A’s in the Treatment of Obesity in a GP’s Practice 

This tool is derived from smoking addiction counseling and was also proposed many years ago for the treatment of obesity.

It has been observed that the use of all elements of the five A rule significantly increases the achievement of therapeutic success.

The use of rule the five A’s include:ASK—asking questions should be a motivational interview. During the interview, the patient should be made aware of the impact of their body weight on general health and quality of life. Avoid embarrassment, guilt, and stigmatization during the conversation. Always use adequate medical vocabulary and emphasize that obesity is a disease that can and should be treated. One should also avoid judging the patient during the interview. However, the assessment of the patient’s readiness for change cannot be avoided.

There are many standardized methods of assessing readiness for change, but in the conditions of everyday clinical practice, it is enough to ask the patient the following five questions:(1)Does the patient want to be treated for obesity to improve their health?(2)Does the patient want to change his or her eating habits permanently and does not see it as a struggle?(3)Does the patient feel that their current way of eating is harmful to them?(4)Is the patient aware that the treatment will be long and is ready to cooperate with their doctor?(5)Will the patient try to accept the proposed treatments?

If the patient is not ready to change, methods should be implemented to motivate them to make the change. In addition, the patient’s sense of self-efficacy should be built by explaining to them that he is not expected to make a complete revolution in their life and that treatment will be based on small, gradual changes.

2.ASSESS—assessment of the causes of weight gain, health status, and occurrence of complications caused by excess fat in the body. It is very important to correctly and fully determine the cause of weight gain, especially emotional eating and eating disorders (BED and NES). The patient’s physical health can be assessed on the basis of a 100-point visual analog scale (VAS). Screening for depression (the Beck scale) and the Hospital Anxiety and Depression Scale (HADS) should also be performed. Anamnesis should also be taken with the patient regarding chronic diseases, and in the absence of a prior diagnosis of obesity complications, their diagnosis should be undertaken.3.ADVICE—presenting treatment options that can be used in a particular patient. In the selection of therapeutic methods, the primary cause of obesity should be considered, followed by the stage of the disease and the occurring complications. It is very important that, during the conversation with the patient about the recommendations, they have a sense of understanding. In addition, the patient should be made aware that the treatment process will be long and requires commitment from them and that the doctor and other members of the therapeutic team are there to help them overcome difficulties. The patient should be presented with all therapeutic options that should be used in their case and discuss the benefits and possible risks associated with them.4.AGREE—obtaining the patient’s consent to the proposed therapeutic goal and treatment plan. It is necessary to be aware that it is the patient who implements the doctor’s recommendations; therefore, they cannot be arbitrary and must consider the patient’s capabilities and the degree to which they are willing to comply with the recommendations. In other words, this stage is a compromise between what the patient should do, according to the doctor, and what the patient can and wants to achieve. At this stage, negotiations should be conducted with the patient based on respect for their autonomy and their right to choose. However, the choice should be conscious, i.e., the consequences should be explained to the patient. Obtaining the patient’s acceptance of the proposed therapeutic goal and treatment plan may require many discussions. This should not discourage the doctor from taking them. In addition, the physician must be willing to modify his recommendations based on the needs and capabilities of the patient.It is very important at this stage to work on realizing the patient’s expectations regarding weight loss. The patient should also be made aware that meeting the behavioral change goals is more important than weight loss itself because this will ultimately help them achieve the intended weight reduction. Success for each patient will have a different dimension, but it is important that the patient focuses on improving mental and physical health, not on the number of kilograms lost.5.ASSIST—supporting the patient in the therapeutic process. After agreeing on their therapeutic goal, the doctor should help the patient identify barriers that may hinder treatment (social, medical, emotional, and economic) and factors that facilitate treatment (motivation and social support). The role of the doctor is to identify the causes of the disease, educate, recommend adequate therapeutic methods, and support the patient in their implementation. An important element of support is setting the schedule of follow-up visits, determining their frequency, and informing the patient what will be checked during the visit, which will make it easier for the patient to implement the recommendations. The schedule should specify the number of visits necessary to achieve the therapeutic goal, minimum and maximum time intervals between visits (the exact date of the next visit should be determined at the previous visit), parameters that will be checked during the visit, and what should be brought to the next visit (e.g., physical activity and results of additional tests).At each follow-up visit, new problems that make it difficult for the patient to comply with the recommendations should be identified, and solutions or other therapeutic methods should be introduced to eliminate the existing problems [66,67,68,69,70,71,72,73].

### 5.3. Nutritional Interventions

The term ‘diet’ defines a way of eating; therefore, everything a person eats is a diet. However, in the common consciousness, diet is associated with a special way of nutrition (elimination of many foods), which—used for several days or weeks—will lead to weight loss body, after which one can eat as before [74].

That is why it is better to talk to the patient not using the word ‘diet’, just to make a permanent change in eating habits.

The energy content of the diet should be determined individually. The simplest is to apply a formula to determine the total energy expenditure.

Basic energy expenditure (basal metabolic rate (BMR) × coefficient physical activity)

BMR:

For men = 11.6 × body weight (kg) + 879 kcal;

For women = 8.7 × body weight (kg) + 826 kcal.

The physical activity factor:

For people who lead a sedentary lifestyle—1.3;

Moderately active—1.5;

Regularly physically active—1.7 [74]. 

From the calculated energy expenditure determining the energy content of the diet, it is necessary to subtract about 500–600 kcal to obtain approximately 0.5 kg weight loss per week or 1000 kcal for a loss of approx. 1 kg per week. Reassessment of the energy content of the diet should be made in accordance with the above data each time the body weight stops reducing [74].

A diet should be varied and contain all the necessary food ingredients. In the selection of recommended foods, individual patients’ preferences should be considered. The proportion of food macronutrients recommended by the WHO is as follows: about 20% of the energy content of the diet should be proteins, about 25% fats, and about 55% carbohydrates [75].

No more than 10% of energy may come from fats containing saturated fatty acids (SFA). At least 6% of this energy should provide polyunsaturated fatty acids (PUFA), and the rest should provide monounsaturated fatty acids with cis configuration (MUFA). It should be noted that monounsaturated fatty acids with the configuration trans (trans fatty acids—TFA) should not exceed 1% of the incoming energy from fats [75].

The main sources of SFA in the diet are butter and lard, beef tallow, as well as oils: coconut and palm, and also cocoa, nut, and vegetable butter (these kinds of butter are the main ingredients of chocolate) [75].

The main food sources of MUFA are olive oil and other vegetable oils [75]. 

TFAs are mainly delivered from fast food, cakes, and cookies that contain industrially hydrogenated vegetable oils included in the composition of shortenings, fries, and margarine [76]. 

Note that in intake, PUFA ω-6 and ω-3 should be maintained at a proper 4:1 ratio. Foods rich in ω-3 fatty acids are herring, tuna, salmon, sardines, mackerel, trout, and oil fish. The main food sources of ω-6 acids (>60%) are oils: soybean, sunflower, safflower, evening primrose, and oils from grape seeds, poppy seeds, borage of medicine, and blackcurrant. Approximately 40–50% of these fatty acids include oil, wheat germ, corn, nuts, walnuts, cottonseed, and sesame [77].

Simple carbohydrates (e.g., glucose, fructose, lactose, xylitol, and sucrose) should provide <10% of energy. Dietary fiber should provide wholegrain bread, other grain products and vegetables, fruits, and plant legumes [78].

Reducing the amount of food alone should not be recommended, but most of all, changing its quality (e.g., consumption of fewer dairy products fat, cooking or roasting meat instead of frying, cooking soups on vegetable stocks, without roux and with yogurt instead sour cream, and not using mayonnaise for salads).

The patient should be made aware from the outset that the changes it introduces must be permanent. However, this does not mean that there are any foods that they will not be able to eat until the end of their life. If they eat a high-energy product very rarely, e.g., once a quarter, this will not cause weight gain.

Regular consumption should be recommended (at similar times) 3–5 meals a day, finishing eating with a feeling of incomplete satiety, eating between meals (in the case of not feeling very hungry, one can drink a glass of water or eat a vegetable, not fruit), not eating food while watching TV or reading or computer use, and coping with stress other than through overeating.

The distribution of energy when eating five meals:

Breakfast—25%;

Second breakfast—15%;

Lunch—35%;

Afternoon tea—10%;

Dinner—15%.

The distribution of energy when eating three meals:

Breakfast—40%;

Lunch—40%;

Dinner—20% [74,79].

Popular ‘miracle diets’ are not recommended. Both high-fat and high-protein diets, with significantly higher-than-recommended amounts of cholesterol, promote the development of atherosclerosis. Moreover, they are ketogenic diets, which on the one hand, have an effect of inhibiting the feeling of hunger, but on the other hand, lead to acidification of the body and electrolyte disorders. High-protein diets also contain higher-than-recommended phosphate content, which causes calcium malabsorption and may develop osteoporosis with prolonged use. Low-energy and very low-energy fat-free diets cause significant weight loss, which promotes the ‘yo-yo’ effect and also has a ketogenic effect [74,80,81,82]. Recently published studies indicate that the use of ‘miracle diets’ is a risk factor for the development of emotional eating and eating disorders [20]. 

### 5.4. Behavioral Therapy

Lifestyle-changing therapy for patients who are overweight or obese should contain behavioral intervention that improves adherence to reducing dietary recommendations energy of meals and affects increased physical activity. Behavioral intervention may include self-control in terms of body weight and consumption of meals and physical activity, clear and precise defining the goals of therapy and education on obesity, nutrition, and physical activity, individual and group conversation, stimulus control, systematic solving emerging problems, reducing stress, cognitive behavioral therapy, motivational interview, behavioral agreement, psychological counseling, and social support mobilization [83,84,85,86].

If the patient fails to achieve a 2.5% reduction in body weight in the first month of treatment, intervention and support should be stepped up to behavioral, as early body weight reduction is a key, long-term indicator of success in losing body weight [73]. 

The GP should discuss with the patient realistic treatment goals. The goal is to lose weight by about 10% in 3–6 months, then maintain this reduced weight for several months, and then act to reduce body weight further.

The family doctor should also explain to the patient that:-Losing weight too quickly is not beneficial for health (risk of developing liver steatosis and gallstones) and is associated with risks such as the ‘yo-yo’ effect (loss of lean mass and lowering the level of basic expenditure energy);-The use of a very restrictive diet may cause deficiencies in vitamins and microelements;-Treatment is not a short period of dieting, but a permanent change in lifestyle, including habits, nutrition, and increasing physical activity, and any unfavorable change in this aspect will lead to disease relapse;-The real success is long-term maintenance weight loss of at least 10% from the initial body weight, not the number of kilograms that the patient will be rid of.

The family doctor should also conduct an analysis of the patient’s eating habits that must be eliminated:-Eating while watching TV;-Calming oneself with food;-Eating foods with the wrong composition;-Eating in a hurry;-Eating under the influence of the greatest hunger;-Eating between meals;-Irregular eating habits.

The GP should advise the patient to keep a food diary for at least 3 months. In the diary, before eating a meal, the patient records the time of consumption, composition, weight, and caloric value. Remember to save as well all fluids consumed except water, unsweetened coffee, and tea. It is also worth recording the patients’ physical activity as possible problems may arise with insufficient lifestyle changes [83,84,85,86].

### 5.5. Physical Activity

Aerobic exercise should be recommended (prescribed) to patients who are overweight and obese as a part of lifestyle intervention. It may be initially advisable to recommend a gradually increased amount and intensity of exercise; ultimately, this should be at least 150 min per week of moderate-intensity exercise divided into 3–5 sessions. 

In the treatment of weight gain and its prevention in a patient implementing the program, 60–90 min of moderate daily exercise in leisure time is recommended for weight loss. 

A dynamic, aerobic effort is recommended, involving large muscle groups. Recommended forms of physical activity for obese adults: brisk walking, cycling, swimming, water exercises, and Nordic walking. 

Resistance exercise should be recommended (prescribed) to patients undergoing an intervention weight loss for supporting loss of body fat while maintaining lean mass; ultimately, these should be single sets of engaging resistance exercises for major muscle groups performed 2–3 times a week. 

In addition to aerobic exercise, the patient should do resistance exercises 2–3 times a week 12–15 repetitions each, with a commitment of 30–50% of maximum muscle strength. 

The target training heart rate should be 60–70% maximum heart rate (220 minus age) in people without cardiovascular disease, and in people with cardiovascular disease, 40–70% heart rate reserve (highest heart rate achieved during the exercise test minus resting heart rate) plus resting heart rate. 

Absolute contraindications to treatment movement are decompensated circulatory failure, unstable coronary artery disease, and respiratory failure.

Carefully, under medical and rehabilitation supervision, physical activity should be recommended in patients with a BMI > 40.

All overweight or obese patients, apart from physical exercise, should be encouraged to spend their free time actively to reduce their sedentary lifestyle. 

In order to improve the engagement for an individual’s plan of activity, the involvement of trained and certified fitness professionals should be considered [74,87,88].

### 5.6. Psychotherapy

All patients diagnosed after screening for depression or anxiety should be referred to a psychologist. 

Indications for patient referral to a psychologist dealing with eating disturbances also include the following:-Emotional eating;-Low self-esteem;-Suspected NES;-Suspected BED;-Suspected food addiction.

The main recommendation is cognitive behavioral therapy (CBT). It is a combination of behavioral therapy (oriented to change behavior) and cognitive trends, referring to the patient’s perception and understanding of the world, their thoughts, beliefs, imagination, and goals. CBT helps the patient to identify and possibly change their own cognitive constructs (concerning, for example, themself, life situation, illness, and future) and shape new behaviors and skills that will be helpful in achieving their assumed goals. 

The beliefs subjected to analysis and modification primarily relate to issues related to obesity, its consequences, and the possibility of regulating body weight. Changing behaviors, in turn, concerns those activities that are directly related to weight loss and maintaining the achieved results. CBT in the treatment of obesity should include the following elements:-Self-monitoring (e.g., keeping a food diary);-Techniques to control the eating process (e.g., slow chewing);-Control of stimuli and their reinforcement or reduction (e.g., shopping according to a list);-Additional cognitive techniques;-Relaxation techniques [89,90].

Another useful treatment of obesity is interpersonal therapy (IPT), which combines elements of cognitive behavioral and psychodynamic approaches (attachment theory). Interpersonal therapy is considered to be particularly effective in treating BED [91]. 

Many studies also confirm the effectiveness of psychodynamic therapy in the treatment of patients with obesity. This trend primarily focuses on early childhood experiences, unconscious drives, internal conflicts, as well as mental defense mechanisms. It aims to thoroughly analyze the mechanisms of the patient’s mental functioning and gain insight due to the reference of subjective tools (e.g., interpretations) and phenomena (e.g., transference and countertransference, free association, dreams) [92,93].

### 5.7. Pharmacotherapy

There is no drug that can cure obesity. Currently, available drugs can only support the treatment of obesity through various mechanisms of action. Therefore, pharmacotherapy for overweight and obesity should be used only as an adjunct to lifestyle therapy and not alone [4]. 

Pharmacotherapy in the treatment of obesity should be used chronically as long as it is effective and well tolerated because obesity is a chronic disease. Short-term pharmacotherapy use (3–6 months) does not cause long-term health benefits and cannot be recommended [4]. Short-term pharmacotherapy use may be associated with short-term weight loss followed by the ‘yo-yo’ effect and negatively affected health [4]. 

The choice of pharmacotherapy should be individual because of the heterogeneity of responses to obesity interventions, including medication [94]. The current standard selection of pharmacotherapy includes physician/patient preference, medication interaction, comorbidities, efficacy, and risk of potential adverse events [94,95,96]. However, new data support the concept that the primary cause of obesity development and the drug’s mechanism of action should be the first criterion for choosing a drug [97]. This approach has already been included in the guidelines of seven Polish Scientific Associations [18] and the Canadian guidelines [98]. 

There are currently four drugs registered in the European Union that help reduce body weight: orlistat, a drug composed of hydrochloride naltrexone and hydrochloride bupropion, and long-acting GLP-1 analogs (liraglutide and semaglutide). 

Pharmacological treatment is indicated in patients with obesity (BMI ≥ 30 kg/m^2^) or overweight (BMI ≥ 27 kg/m^2^) with ≥1 complication of obesity in a patient in whom non-pharmacological treatment has failed to achieve the therapeutic goal [74]. 

Pharmacotherapy can also be used at the stage of maintaining the effects achieved over time with non-pharmacological treatment and after surgical treatment of obesity [74].

If, after 3 months of using pharmacotherapy, weight loss is less than 5% in patients without a diagnosis of type 2 diabetes and less than 3% in people diagnosed with this disease (counting weight loss from drug application), its continuation is unjustified. However, it should be stressed that if the use of pharmacotherapy had no effect, do not wait until 3 months have passed but discuss the implementation of recommended diet and physical activity. In addition, psychological problems should be analyzed, and the use of the prescribed pharmacotherapy checked [74]. 

**Orlistat** (tetrahydrolipstatin, a derivative of lipostatin produced by *Streptomyces toxytricini*) is used orally at a dose of 120 mg three times a day before main meals. In randomized trials, taking orlistat for one year resulted in a weight loss of ~3 kg more than in the placebo group. This drug inhibits the activity of lipases in the gastrointestinal tract: gastric, pancreatic, and intestinal, and prevents the digestion and absorption of some of the fats taken with food. It does not affect the feeling of satiety, hunger, or appetite. It is absorbed from the gastrointestinal tract in trace amounts (1% of the dose), and its metabolites are inactive; therefore, it has no systemic effect. The use of orlistat is justified only in people who prefer fatty foods and have problems with modifying eating habits and who are aware of the drug’s mechanism of action and possible side effects. Consumption of food that contains too many fats results in increased frequency of bowel movements, loose and liquid stools, fatty stools, an urgency to defecate, fecal incontinence, bloating, and abdominal pain. The patient should be warned that these are the effects of nutritional errors, and reducing the consumption of fats will eliminate their occurrence. Patients using lipophilic drugs should wait for ≥2 h between taking them and using orlistat. Contraindications are hypersensitivity to the drug, pregnancy and lactation, cholestasis, and chronic malabsorption syndromes [99,100,101,102,103]. 

Indications for the use of orlistat:Obesity (BMI ≥ 30 kg/m^2^);Overweight (BMI ≥ 27 kg/m^2^), with obesity complications, such as hypertension, lipid disturbances, ischemic disease, myocardial infarction, type 2 diabetes, sleep apnea, or PCOS.

Contraindications to the use of orlistat:Chronic malabsorption syndrome;Cholestasis;Pregnancy;Breast-feeding;Hypersensitivity to orlistat.

**Combined preparation containing two active substances, naltrexone hydrochloride and bupropion hydrochloride,** is in one tablet. The prolonged-release tablet contains 7.2 mg of naltrexone and 78 mg of bupropion (equivalent to 8 mg of naltrexone hydrochloride and 90 mg of bupropion hydrochloride). Treatment begins with taking one tablet in the morning for a week, then the next week—one tablet in the morning and one in the evening. In the next two tablets, one in the morning and one in the evening, and in the 4th week, the target dose is introduced as two tablets, one in the morning and two in the evening (the daily target dose is 28.8 mg naltrexone and 312 mg bupropion, equivalent to 32 mg naltrexone hydrochloride and 360 mg bupropion hydrochloride). If, after 16 weeks of using the preparation, the patient’s body weight has not decreased by ≥5% of the initial value, the drug should be discontinued [104]. 

Naltrexone and bupropion act on the same regions of the central nervous system (the arcuate nucleus of the hypothalamus and the mesolimbic dopaminergic reward system), and the combination has a hyperadditive effect on the regulation of food intake. This allows them to be used in lower doses, which reduces the risk of side effects and promotes better tolerance of treatment [104]. 

Bupropion is a dopamine and norepinephrine reuptake inhibitor (NDRI) and a non-competitive nicotinic receptor antagonist of the β-ketoamphetamine class. It is used alone to treat depression, seasonal affective disorder, and nicotine addiction. Naltrexone, in turn, is an antagonist of the µ-opioid receptor, to a lesser extent of the κ-receptor, and to an even lesser extent of the γ-receptor. At a dose of 50 mg, it is used in the treatment of non-opioid addictions, primarily from alcohol (supporting abstinence by reducing the need to drink) [104].

Bupropion in the arcuate nucleus of the hypothalamus stimulates the activity of POMC-secreting neurons and, as β-ketoamphetamine, the release of cocaine- and amphetamine-regulated transcript (CART), which in turn stimulates the release of α-melanocortin (α-MSH), which binds to melanocortin type 4 receptors (MC4-R), stimulating the feeling of satiety. The feedback that inhibits the release of POMC is the increase in the release of β-endorphin by this neurotransmitter; however, naltrexone—by blocking the µ-opioid receptors—inhibits the feedback loop and, as a result, prolongs the feeling of satiety. Naltrexone and bupropion also reduce food intake stimulated by appetite (the hedonistic search for a specific food not to satisfy hunger but to feel pleasure from its consumption), which is the responsibility of the reward system along with its main neurotransmitters: dopamine, norepinephrine, and endogenous opioids. Bupropion inhibits the reuptake of dopamine and noradrenaline (inhibition of the drive to seek food), and naltrexone blocks opioid receptors, which stimulates the secretion of endogenous opioids (reduces the ‘liking’ of tasty food) [104].

In clinical trials, the most common adverse drug reactions to this combination product were nausea, vomiting, headache, dizziness, insomnia, and dry mouth. They usually spontaneously disappear within the first 4 weeks of treatment [105,106,107,108].

Indications for the use of the drug include supplementing with this drug a diet with reduced energy content with increased physical activity for weight loss in adult patients (≥18 years old) with a baseline BMI of either:BMI ≥ 30 kg/m^2^ (obesity);BMI 27 kg/m^2^ to <30 kg/m^2^ (overweight) if the patient has one or more complications of obesity (e.g., type 2 diabetes, dyslipidemia, compensated hypertension).

Contraindications:Hypersensitivity to any substance active or auxiliary;Uncontrolled high blood pressure;Current epilepsy or seizures in the interview;A tumor of the central nervous system;The period immediately following an abrupt withdrawal from alcohol or benzodiazepines in an addicted person;History of bipolar disorder;Taking bupropion or naltrexone for another indication other than weight loss;Bulimia nervosa or anorexia nervosa now or in the past;Addiction to long-term use of opioids or opiates (e.g., methadone) and shortly after their discontinuation in the addicted person;Taking monoamine oxidase inhibitors (MAOI);Severe liver problems;End-stage renal failure or severe disorders of kidney function;Pregnancy and breastfeeding.

The use of this drug as the first choice is recommended in patients diagnosed with emotional eating, BED, NES, food addiction, depression, and smoking cessation [18,100].

**Liraglutide** is a long-acting GLP-1 analog that is used s.c. 1 × daily at the target dose of 3 mg/d. Treatment of obesity is started with a dose of 0.6 mg/d and increased weekly by 0.6 mg/d until a dose of 3 mg/d is reached. If the drug is poorly tolerated for another 2 weeks after increasing the dose, discontinuation of the drug should be considered. Treatment should be discontinued if, after 12 weeks of use at a dose of 3.0 mg/d, the patient has not lost ≥5% of the initial body weight.

Liraglutide, such as natural GLP-1, affects target cells, producing effects analogous to the natural hormone. The main mechanism leading to weight loss depends on the direct activation of GLP-1 receptors located in the central nervous system and the downstream activation of GLP-1 afferents, including neurons of the autonomic nervous system. GLP-1 receptors are found in many structures of the central nervous system, including solitary tract nuclei and POMC/CART anorexigenic neurons of the hypothalamus, and their activation is responsible for the feeling of satiety. The concomitant inhibition of hunger is the result of indirect inhibition of neurotransmission in NPY- and AgRP-expressing neurons through γ-aminobutyric acid (GABA)-dependent signaling. The additional mechanism of increased satiety is slowing down gastric emptying [109]. Experimental studies in rats also indicate that reduced food intake may be related to nausea, which is induced by the effect of liraglutide on GLP-1 receptors in the solitary tract nucleus [110,111,112,113].

Liraglutide also acts in many peripheral tissues. The incretin effect exerted by GLP-1 agonists was the first to be discovered, including GLP-1-stimulated increased glucose-dependent insulin secretion from pancreatic β-cells, which is used in the treatment of type 2 diabetes, where the target dose is 1.8 mg/d (liraglutide has been approved under a different trade name for the treatment of diabetes) [109]. Based on a trial performed on patients with type 2 diabetes and treated with liraglutide at a dose of 1.8 mg/d, this treatment did not increase the risk of cardiovascular complications [114]. However, there are no prospective clinical trials conducted in patients with obesity but without type 2 diabetes and treated with liraglutide in a dose of 3.0 mg/d. The improvement of cardiometabolic parameters in people without type 2 diabetes primarily depends on the reduction of body weight.

Indications are similar to indications for the use of other drugs supporting the treatment of obesity; the use of liraglutide 3 mg in the treatment of overweight and obesity should be considered as an adjunct to lifestyle modification in patients:(1)With a BMI ≥ 30 kg/m^2^ (obesity);(2)With a BMI of 27–30 kg/m^2^ (overweight) if accompanied by ≥1 of complications related to excessive body weight (including prediabetes or type 2 diabetes, hypertension, lipid disorders, or obstructive sleep apnea).

The effectiveness of treatment is assessed after 12 weeks of using liraglutide in a full dose of 3.0 mg 1 × daily s.c.; it may be continued if body weight has decreased by ≥5%. The most common side effects are nausea, vomiting, diarrhea, and constipation, which are usually temporary [115,116,117].

This drug should be the first choice in the treatment of obesity in patients with prediabetes or type 2 diabetes, as well as clinical features of insulin resistance after the exclusion of emotional eating, BED, food addiction, and NES [17,97]. Studies conducted using functional magnetic resonance imaging confirmed the lack of influence of GLP-1 analogs on the reward system [118] and their lower effectiveness in people with emotional eating to stimulate the reward system [119]. 

Contraindications for the use of liraglutide, apart from hypersensitivity to the active substance or excipients, include a family history of medullary thyroid cancer, a history of pancreatitis, and pregnancy. 

**Semaglutide** is a very long-acting GLP-1 analog that is used s.c. once a week at a target dose of 2.4 mg/week. Treatment of obesity starts with a dose of 0.25 mg/week. After 4 weeks, it is increased to 0.5 mg/week, after another 4 weeks to 1 mg/week, after another 4 weeks to 1.7 mg/week, and a further 4 weeks to the target dose of 2.4 mg/week. If the drug is poorly tolerated 2 weeks after increasing the dose, discontinuation should be considered. If severe gastrointestinal symptoms occur, consideration should be given to delaying dose escalation or reverting to the previous dose until symptoms have improved. Due to the long half-life, the drug should be discontinued 2 months before the planned pregnancy.

The mechanism of action of semaglutide is similar to that of liraglutide [109]. Similar to liraglutide was originally registered for the treatment of type 2 diabetes, where the target dose is 1 mg/week (for the treatment of diabetes, semaglutide has been registered under a different trade name). To date, no prospective clinical trials have been conducted to evaluate the effect of semaglutide on cardiovascular risk in non-diabetic subjects. The improvement of cardiometabolic parameters in people without type 2 diabetes primarily depends on the reduction of body weight.

Indications are similar to those for the use of other drugs supporting the treatment of obesity; the use of semaglutide 2.4 mg in the treatment of overweight and obesity should be considered as an adjunct to lifestyle modification in patients:(1)With a BMI ≥ 30 kg/m^2^ (obesity);(2)With a BMI of 27–30 kg/m^2^ (overweight) if accompanied by ≥1 of complications related to excessive body weight (including prediabetes or type 2 diabetes, hypertension, lipid disorders, obstructive sleep apnea, or cardiovascular disease).

The most common adverse drug reactions are nausea, vomiting, diarrhea, constipation, and abdominal pain, which are usually temporary. Due to the rapid initial weight loss, there is also a risk of developing gallstones [120,121,122,123].

Contraindications to the use of semaglutide, apart from hypersensitivity to the active substance or excipients, include a family history of medullary thyroid cancer, history of pancreatitis, and pregnancy. 

If patients have psychogenic eating disorders, pharmacotherapy with liraglutide and semaglutide may be less effective, and it is suggested that a combination of naltrexone and bupropion should be considered first. Some authors also propose considering the use of polytherapy with liraglutide and naltrexone with bupropion [20]. The safety of the combination of naltrexone/bupropion and a long-acting GLP-1 analog was confirmed in the recently published post hoc analysis of the LIGHT study [124].

It should be noted that a systematic review and meta-analysis of randomized placebo-controlled trials showed that the use of GLP-1 analogs is associated with an increased risk of developing gallstones or cholelithiasis, and this risk increases with higher doses, longer duration of use, and use for weight loss [125]. In addition, the analysis of cases reported in the European Pharmacovigilance Database showed that the use of GLP-1 analogs is associated with a higher risk of developing thyroid cancer [126]. In a recently published study, which analyzed a total of 2526 cases of patients with thyroid cancer compared with 45,184 people from the control group, it was shown that the use of GLP-1 analogs for 1–3 years was associated with an increased risk of all thyroid cancers [127]. Attention should also be paid to the increased risk of tachycardia and arrhythmia during treatment with semaglutide [128]. 

According to the ESE guidelines from 2020, it is not recommended to use metformin solely for weight reduction (no registration in this indication), and indications for the use of this drug are prediabetes and type 2 diabetes [22].

**Other drugs:** there is insufficient evidence to support the use of herbal medicines, dietary supplements, probiotics, or homeopathy in the treatment of obesity. The results of single studies indicate that the use of fiber preparations containing soluble and insoluble fibers may increase the effects of non-pharmacological treatment of obesity.

### 5.8. Bariatric Surgery

#### 5.8.1. Requirements for Reference Centers

Bariatric surgeries should be performed in centers specializing in this type of surgery, able to choose the optimal (medical indications and patient’s preferences) surgical method together with the patient, having substantive preparation and appropriate equipment. This is not only the equipment of the operating room (e.g., operating table and laparoscopic tools) but also the equipment of the ward (hospital beds with a load capacity of 250–300 kg, couches, wheelchairs, chairs, and bariatric platforms) and sanitary facilities (shower cabins adapted for people with obesity equipped with appropriate handles and handrails) [129].

#### 5.8.2. Qualification

Proper qualification for surgical treatment of obesity is one of the key factors affecting its results. It is recommended that non-surgical treatment should be attempted before considering surgery [130].

The primary criterion for qualifying for bariatric surgery is the patient’s BMI. Until recently, the second crucial criterion assessed during qualification was the patient’s age [130]. Currently, there is no age limit for patients undergoing bariatric surgery; however, a careful selection of older patients is recommended, in whom frailty assessment is critical, which, more than age, is associated with a higher rate of postoperative complications. 

It is worth noting that before surgery, it is recommended to lose 5–10% of body weight (among others, it has a positive effect on the results of bariatric treatment and reduces the risk of perioperative complications) [131]. 

Eligibility criteria based on BMI:BMI > 40.0 kg/m^2^;BMI 35.0–39.9 kg/m^2^ in a patient with obesity complications (e.g., type 2 diabetes, hypertension, severe joint diseases, dyslipidemia, a severe form of OSA). The latest guidelines recommend surgery in patients with BMI in this range, regardless of obesity complications;BMI 30.0–35.0 kg/m^2^ and uncontrolled type 2 diabetes despite appropriate pharmacological treatment [130,131].

Contraindications to bariatric procedures:Mental disorders—personality disorders, severe depression;Alcoholism;Drug abuse;Eating disorders;No possibility of proper, long-term postoperative care;Poor long-term prognosis due to life-threatening diseases [132].

The final qualification of the patient for the operation must be multidisciplinary. It is a decision of a team of specialists experienced in the treatment of obesity: a surgeon, an internist, an anesthesiologist, a psychologist or a psychiatrist, a dietitian, a physiotherapist, and, if necessary, a cardiologist, a pulmonologist, a gastroenterologist, and a neurologist [129]. 

The optimal time to prepare the patient for surgery should not be shorter than three months but longer than 6–12 months [132]. 

The preoperative assessment of patients scheduled for bariatric surgery should be much broader than any other major abdominal surgery. The success of surgical treatment of obesity requires a good understanding of the entire treatment process by the patient, not only the surgery itself. Therefore, the patient must be provided with information on the benefits and consequences and the risks associated with the operation.

A key, unfortunately often overlooked, aspect when qualifying a patient for surgery is a psychological consultation (comprehensive assessment sometimes requires several meetings with a psychologist), which includes behavioral, nutritional, and family psychological assessment, and personality factors should be an integral part of the patient’s preoperative assessment [133]. Working with a psychologist is one of the elements aimed at increasing the safety and effectiveness of surgical treatment by identifying specific areas to create an individually tailored treatment plan.

#### 5.8.3. Types of Operation

There are at least a dozen different types of operations to choose from, more or less changing or modifying not only the anatomy but also the physiology of the digestive system, and characterized by a different number and type of long-term complications. The common feature of all surgical methods is the preferred access method—laparoscopy 2D or 3D [134,135]. Detailed qualification criteria for a given type of bariatric operation go beyond the thematic framework of this study. It should be emphasized that bariatric treatment is personalized, and each element (mainly surgical procedures) should be individually selected for the patient. 

#### 5.8.4. Post-Treatment Monitoring and Intervention

##### In-Hospital

Perioperative care for patients undergoing bariatric treatment should be organized according to the Enhanced Recovery After Bariatric Surgery principles [136,137]. Patients with morbid obesity have an increased risk of partial atelectasis in the distal alveoli. In the postoperative period, in the recovery room, it is recommended to administer supplemental oxygen [138]. For patients diagnosed with obstructive sleep apnea, it is necessary to use CPAP, i.e., breathing support that prevents the collapse of the alveoli. In the postoperative period, an important aspect is appropriate respiratory rehabilitation and breathing exercises. Each patient should be mobilized on the first day after returning from the post-operative observation unit. Patients can drink fluids after returning from the recovery room [139]. On the first postoperative day, the patient orally takes fluids (no daily volume limit) and oral nutritional support (ONS). Patients are encouraged to drink fluids while walking around the hospital ward (simultaneous activation). In the following days, gradually expand the diet. Each patient receives proton pump inhibitors (PPI) twice a day, antithrombotic prophylaxis, and non-opioid analgesia until discharge. Discharge occurs on either postoperative day 1 or 2 upon meeting specific discharge criteria:Patient tolerates oral diet and drinks at least 1000 mL of fluids per day;Does not require intravenous fluids;Postoperative pain is manageable with oral medication;The level of physical activity is similar to that before the operation;After discharge, the patient will remain under the care of third parties and, if necessary, contact with the treatment center is ensured;There were no complications that required hospitalization [132].

Upon discharge, patients receive a follow-up visit plan for 1 year, a baseline dietary plan, and a prescription for PPI, antithrombotic prophylaxis, vitamin supplementation, and ursodeoxycholic acid (gallstone prevention) [140].

##### Perioperative Monitoring for up to 30 Days

The wound should be kept clean and washed every day with a disinfectant and dressing applied. Stitches should be removed 7–10 days after the surgery; for patients with diabetes, this time may be longer. It can be performed by a primary care physician or in a surgical outpatient clinic. Patients with fever, vomiting, wound discharge, abdominal pain requiring opioids, and dehydration symptoms should be referred to a surgical unit. A follow-up visit with a surgeon is necessary during the first 30 days. 

Every patient is prescribed low-molecule heparin injections. There are no clear standards for how long the antithrombotic prophylaxis should be administered; however, it should not be shorter than 7 days after discharge [132]. PPI should be taken twice a day for 30 days. A longer period should be considered in patients with symptoms of gastroesophageal reflux after surgery. Rapid weight loss is associated with an increased risk of the development of gallstones. Recently, several randomized and observational studies have shown that the postoperative supply of ursodeoxycholic acid significantly reduces the risk of the development of biliary stones. The debate regarding the duration of such prophylaxis and the dose is still ongoing; however, at the moment, it is advisable to consider the use of 500 mg ursodeoxycholic acid in a divided dose for 6 months [137].

From 2–4 weeks, the patient can take solid foods, depending on tolerance. Quick and effective activation of the gastrointestinal tract after bariatric surgery would not be possible without proper patient education in the preoperative period. In the perioperative and postoperative periods, the patient should also be consulted by a dietitian who will coordinate further nutrition of the patient. Due to the initial difficulty in taking food, patients need to have an adequate protein supply of about 1–1.5 g of protein per kilogram of ideal body weight. Initially, the most important aspect of nutrition for patients after surgery is an adequate amount of fluids to avoid dehydration. An operation with a large malabsorptive effect may require an even greater supply of protein. Each patient, during the first 30 days, should have a follow-up visit with a dietitian to assess their diet plan.

Patients should resume physical activity straight after surgery. For 6 weeks, patients should not perform exercises involving the abdomen. Preferably, physical activity should be planned and supervised by a physiotherapist. 

##### Monitoring during the First Year after Surgery

Patients after bariatric surgery require follow-up visits to determine weight loss, remission of obesity complications, complications associated with surgery, assessment of nutritional status, and potential qualification for revisionary surgery. The success of weight loss should be assessed as a % of excessive weight loss (%EWL). More than 50% EWL is considered a success. Obesity complications, such as type 2 diabetes and hypertension, need to be closely monitored by primary care physicians or specialists to reduce the dose of medication at a proper time. The remission should be assessed according to treatment standards for specific diseases. Female patients of childbearing age should be advised to be on medically effective contraception for 12 months following the surgery—pregnancy during the time of weight loss is not recommended. 

Malnutrition after bariatric surgery is common. It partly depends on the type of surgery performed (it is more common after RYGB than SG) and also on the initial nutritional status of the patient. People with obesity are usually deficient in vitamins, particularly vitamin D, B_12_, thiamine, and folic acid, as well as calcium, iron, zinc, and copper. In recent years, easily absorbable products for bariatric patients have appeared on the market, which contain all the necessary ingredients [138,139]. In addition, it is recommended to periodically check the level of micro- and macro-elements in the serum. Control tests, including kidney and liver function tests, complete blood count (CBC), and serum ferritin, folic acid, vitamin B_12_, vitamin D, and calcium measurements, should be performed 3, 6, and 12 months after the procedure and then at least once a year [141]. 

If this has not been performed before surgery, intact parathyroid hormone levels should be checked. Serum levels of vitamin A, vitamin E, vitamin K_1_, and PIVKA-II (a protein caused by vitamin K deficiency or antagonism) should be regularly measured at regular intervals after malabsorption procedures, such as BPD/DS, or when symptoms of deficiency occur. It is recommended to monitor serum zinc, copper, and selenium levels after sleeve gastrectomy (SG), Roux-en-Y gastric bypass (RYGB), or BPD/DS. Routine monitoring of magnesium levels is not required.

The patient should be monitored by a ‘bariatric team’, which comprises of surgeon, psychologist, dietitian, and others. The optimal time for a psychologist consultation is between 6 months and 12 months after surgery; at this time, the pace of weight loss slows down, and psychological support enhances the patients’ motivation. A follow-up visit with a surgeon should be scheduled 1 year after the procedure to assess weight, comorbidities, and additional blood tests. Each patient requires a mandatory gastroscopy 1 year after surgery [130].

##### Long-Term Follow-Up

The gastrointestinal tract in bariatric patients’ is permanently changed; thus, they require monitoring indefinitely to determine any signs of malnutrition, macro- and micro-element deficiency, weight regain, or complications, such as gastroesophageal reflux and dumping syndrome. 

Patients with weight regain or onset of surgery-related complications should be referred to a bariatric surgeon to determine the need for revisional surgery. Currently, it is possible to help patients using pharmacotherapy [97]. Pharmacotherapy should be used in patients with a lack of weight loss and as a first-line treatment in patients with weight gain. Currently, available medications are glucagon-like peptide-1 receptor agonist (GLP-1), including liraglutide (3 mg once a day s.c.) and semaglutide (1.5 mg once a week, s.c.) and combination preparation containing naltrexone and bupropion (16 mg/180 mg twice a day, orally). The choice of pharmacotherapy should be individualized. 

Some patients, after massive weight loss, may require plastic surgery to remove excess skin. This operation should be considered at least 12 months after the surgery with at least a 6-month period of stable weight.

### 5.9. Effectiveness of Obesity Treatment

#### Long-Term Monitoring

Beneficial prognostic factors during treatment that indicate long-term maintenance of the effects include a greater frequency of self-monitoring of energy intake and body weight, consistency between dietary choices and weight reduction goals, lower intensity of negative mood, lower intensity of hunger and emotional eating, and regular physical activity. Therefore, it is very important to focus work with the patient on these factors during therapy. The patient should be motivated to self-monitor and make appropriate food choices, which the patient should learn with the support of a dietician and physician. A very important element regarding food choices is directing the patient’s thinking in such a way that there are no dishes and products that are completely forbidden, but there are products that can be eaten in small amounts. The patient must not think that eating a certain product is a sin or a certain product is a ‘forbidden fruit’ because this may become the reason for obsessive thinking about it. Often because of such thinking, the end of the active phase of treatment, i.e., achieving the therapeutic goal for the patient, is the limit beyond which they will be able to eat these forbidden products [142,143,144].

Other very important elements that positively affect long-term weight loss maintenance are regular meals, eating breakfast, and choosing food with a low content of fats and simple sugars.

Long-term weight maintenance is also associated with the implementation of regular physical activity accepted by the patients, and the time of it should be longer than in the phase of active treatment, as lower weight is related to decreased energy expenditure during physical activity [4]. 

There is also a need to effectively treat negative emotions, intensification of hunger, and emotional eating. Both pharmacotherapy and psychotherapy should be used, and, if necessary, both of these therapeutic methods should also be used in the phase of maintaining the achieved treatment effects [4]. 

Numerous data indicate that the psychological assessment of the patient’s personality is prognostic and helps to select those patients who may have problems with maintaining the therapeutic effects and who should be subject to greater supervision at this stage. A stable personality with a higher level of self-control and greater emotional maturity, as well as personality traits such as creativity, autonomy, and self-sufficiency, and an internal locus of control with a sense of self-efficacy and better self-esteem, are favorable prognostic factors. Whereas dysfunctions in social interactions, higher levels of anxiety, and avoidance of monotony are unfavorable prognostic factors [145,146]. 

Maintaining the achieved effects is easier in patients with a more stable living situation and fewer stressful life events and receiving social support. It should be remembered that patients with more problems related to eating behaviors, life situations, and worse functioning, as well as less social support, require more intensive supervision and support and the search for individual solutions in the maintenance phase [145,146].

It is also very important to teach the patient that if the disease recurs, they should immediately contact the doctor and apply the proposed treatment methods. Numerous studies have shown that prolonged therapeutic interventions and continuous professional support in the maintenance phase significantly improve long-term treatment outcomes [145,146].

## 6. Barriers and Overcoming Them in the Treatment of Obesity at the Primary Care Level

The importance of primary care in the management of an overweight or obese patient cannot be overestimated. Family doctors who take care of their patients on a continuous basis, comprehensively assessing their health problems, have a special opportunity to properly prevent, diagnose, and treat people with overweight and obesity. Table 7 summarizes the tasks of a family doctor in this area. 

Due to the fact that primary health care is subject to numerous, varied duties and tasks, the treatment of patients with obesity in the practice of a family doctor may encounter various barriers and difficulties (Table 8).

The basic barrier on the doctor’s side is the lack of knowledge about obesity. Some doctors treat obesity only as a risk factor and not as a complex disease that is the starting point for many organ complications and other diseases. Moreover, doctors have insufficient knowledge about the causes of obesity, its diagnosis, and treatment. Due to insufficient knowledge, but also a misconception, some doctors believe that obesity is the result of a lack of discipline in nutrition, and therefore it is not possible to obtain positive treatment effects. 

The key barrier for most family doctors is certainly the time that the doctor has for the patient. On average, 10 min is allocated for consultation in Polish primary health care. At this time, it is necessary to collect the history, conduct a physical examination, make recommendations, and fill up the patient’s record. The amount of work and short time for consultations discourage doctors from making efforts in the prevention and treatment of obesity in the context of the belief that such activities are of little effectiveness. In addition, there is an inadequate organization of work, in which the role of nurses is omitted, especially at the stage of overweight and obesity diagnosis in relation to anthropometric measurements, but also nurses or other professional personnel in the field of education in the area of obesity prevention and treatment. 

In the effective treatment of a patient with obesity, the patient’s attitude towards this disease and trust in the attending physician is very important. Recently in the mass media, television, and the Internet, the phenomenon of body shaming has been observed. As a result, patients with obesity accept this condition and deny the need for treatment. In addition, a large proportion of patients, as well as some doctors, do not believe in the success and effectiveness of obesity treatment. This reduces the motivation for obesity treatment. This is a serious barrier, and we should only treat patients who consent to it and are sufficiently convinced that it makes sense. On the other hand, in patients already treated, we may encounter serious difficulties in obtaining good results due to family habits and undiagnosed emotional eating or eating disorders. In everyday clinical practice, we observe the occurrence of obesity from generation to generation. It should be emphasized that it is extremely rare that this has a genetic basis. Most often, it is the result of established eating habits and family dysfunction leading to the development of eating disorders. Difficulties in changing eating habits may also be the nature of the work performed (e.g., working as a truck driver or shift work). Excessive eating is also a consequence of the way societies function in the modern world, dominated by haste, stress, competition, irregular, excessive working hours, and the lack of or inactive rest resulting in dealing with negative emotions with food. We are also dealing with unpredictable situations, such as a pandemic that limits people’s activity and worsens their mental state, which contributes to the growing obesity incidence. Both individual and environmental factors are difficult to overcome obstacles in the treatment process. There are also economic reasons. Modern drugs that can be used in the treatment of obesity are expensive, and for the time being, fewer patients can benefit from them. 

Finally, family physicians also encounter some significant barriers at the system level. Due to the complexity of obesity, proper management of it requires a team of professionals, including a dietician, a psychotherapist, and a physiotherapist. It is equally important that in a situation when a family doctor exhausts his possibilities at the level of primary health care, they cannot refer a patient with obesity to a center that will take care of them comprehensively. Therefore, family physicians usually refer patients with obesity to an endocrinologist, surgeon, or other specialists, depending on the occurring complications. 

We are able to overcome some of the above-mentioned barriers. First of all, as doctors, we can broaden our knowledge about diagnostics and treatment of obesity and use the available guidelines on this topic. We are also able to change our attitude and approach to obesity treatment, improve communication with patients, motivate patients to start treatment and persevere in changing their lifestyles. However, it is a lengthy process, requiring consistency and commitment on the part of the doctor, but also patients’ cooperation. Organizational changes can help us with this. One can use the organizational scheme of conduct—different in the case of newly enrolled patients to the practice and different in the case of patients already under our care, which is illustrated in Table 8. The key seems to be the obligation imposed on primary care physicians by the National Health in the year of anthropometric measurements in all patients. This allows the diagnosis of excessive body weight, which is the starting point for further action [74]. 

The second important barrier to overcome is the time of medical consultation. Due to the short time that a family doctor can devote to a patient during one visit, it is beneficial to schedule several short visits in succession, during which the necessary activities related to the diagnosis and treatment of obesity are performed. A randomized study has shown that even a very short, 30 s medical consultation in a primary care clinic and referral of the patient to participate in a local weight loss program results in greater weight loss assessed after 12 months [147]. Certain opportunities in overcoming the above-mentioned barriers also open with the introduction of coordinated care in primary health care in relation to cardiological, endocrinological, pulmonary diseases, and diabetes. In coordinated care, family doctors receive dietary and educational advice as well as greater diagnostic possibilities, which should result in better care for patients with obesity.

## 7. Recommendations 

### 7.1. Recommendations for General Practitioners (GPs)

Screening for overweight and obesity, including weight measurement and BMI calculation, should be performed on all adult patients reporting to their GP once a year;The measurements of body weight, height, and waist circumference should be an integral part of physical examination and should be recorded in the medical history.

The measurements should be performed during the following:

The first visit of patients in GP (at the latest during two consecutive visits);Patient visit due to overweight and obesity;If possible, at each visit the reason for which are complications of obesity, including hypertension, type 2 diabetes, dyslipidemia, coronary artery disease, osteoarthritis, and other comorbidities related to obesity;At each routine visit, if a doctor suspects a patient is overweight or obese;In all patients with normal BMI values (18.5–24.9 kg/m^2^), waist circumference should be measured to assess metabolic risk;In all patients with BMI < 35 kg/m^2^, waist circumference should be measured to assess visceral obesity occurrence;In all patients with overweight and obesity, anamnesis should be taken in the direction of complications, and diagnostics should be performed in their direction. Such activities should be carried out systematically;Diagnostics for overweight and obesity should be performed in all patients treated for their complications;All patients with overweight and obesity should be screened for emotional eating, eating disorders (binge eating syndrome and night eating syndrome), as well as depression and anxiety (HADS);A patient with obesity should be treated with respect, and his/her illness should not be a source of shame and self-blame;After making a diagnosis of overweight or obesity, the doctor should explain to the patient the essence of the disease and its consequences and assess his/her readiness to change and the primary cause of the development of obesity;A physician should use appropriate medical vocabulary in relation to an obese patient, show empathy towards him/her and give advice appropriate to his/her situation, as well as implement all possible therapeutic procedures, including pharmacotherapy and psychotherapy and, if indicated, also surgical treatment. The patient must agree to the proposed treatment methods and accept them;The principle of person-centered care should be the norm in the approach to patients with obesity;During treatment, a schedule of follow-up visits should be set, and the patient should be informed about what will be checked during them. If necessary, expand the methods of implemented treatment and support the patient in the event of difficulties;Remember that a patient with obesity may be aware of their disease, but they do not talk about it because they are ashamed, and the doctor must be able to talk about it;It is unethical not to recognize and not treat obesity instead or refer the patient to another doctor who will treat it.

### 7.2. Recommendations for National Authorities

The lack of designated support for health care of patients with obesity in Poland necessitates the creation of a system based on an obesitologist, a dietician, a psychologist, and a physiotherapist to support general practitioners referring patients with difficulty in diagnosis and management. The already existing shortage of time in primary care precludes taking additional demanding duties, regardless of the transfer of funds for the creation of jobs for other members of the therapeutic team (a dietitian and a psychologist). In order to implement professional treatment of obesity, it is necessary to establish a subspecialty in obesitology, the program of which could be implemented by doctors specializing in internal medicine, family medicine, pediatrics, and general surgery. On the basis of academic centers, regional multi-specialty centers with a full range of care, including the possibility of surgical treatment and pre- and post-operative care (third-level referral centers), should be established. Specialist centers should be established at least voivodeship cities level to provide a full range of conservative treatment (II degree of reference), and selected practices of a family doctor with at least one obesitology specialist in the team should constitute the I degree of reference.

## Figures and Tables

**Table 1 nutrients-15-01641-t001:** Cut-off points of BMI for diagnosis of overweight and obesity in adults according to WHO (1998) and AACE and ACE (2016) [3,4].

WHO (1998)		AACE and ACE (2016)		
	BMI (kg/m^2^)		BMI (kg/m^2^)	Obesity Complications Listed below This Table
Overweight	25.0–29.9	Overweight grade 0	25.0–29.9	None
Obesity grade I	30.0–34.9	Obesity grade 0	≥30	None
Obesity grade II	35.0–39.9	Obesity grade 1	≥25	At least one mild or moderate
Obesity grade III	≥40	Obesity grade 2	≥25	At least one severe

List of complications: pre-diabetes (impaired fasting glucose or/and impaired glucose tolerance), type 2 diabetes mellitus, dyslipidemia, hypertension, cardiovascular disease, non-alcoholic fatty liver disease, polycystic ovary syndrome, fertility disturbances in women, hypogonadism in men, asthma, sleep apnea syndrome, hypoventilation syndrome, gastroesophageal reflux, stress urinary incontinence, osteoarthritis, and depression.

**Table 2 nutrients-15-01641-t002:** A screening tool for diagnosing EE [18].

The Type of Emotions Causing Food Craving	Ask the Patient
Reaching for food during or after a stressful situation caused by both positive and negative factors	Do you feel stomach suction in stressful or anxious situations? Does stress make you reach for food?
Eating when feeling anxious	Do you feel like eating after a stressful situation?
Rewarding oneself with food	Is success food?
Eating when something has failed—comforting with food	When something has not turned out, do you reach for food?
Eating in situations of boredom	When you are bored, do you reach for food? Do you use food during other activities, e.g., reading, watching TV, or working? Do you reach for food while using the computer?
Eating in order to reduce the feeling of fatigue	When you feel tired, does eating help to reduce this feeling?

**Table 3 nutrients-15-01641-t003:** Diagnostic criteria for BED, food addiction, and NES [19,21].

	BED	Food Addiction	NES
Main criterion	Repeated episodes of unrestrained eating at least once a week for three months		Eating a minimum of 25% of the daily food ration after an evening meal or at night with awareness at least twice a week for at least 3 months
Symptoms	At least three of the following symptoms: - Eating much faster than normal; - Eating until feeling uncomfortably full; - Eating large amounts of food without feeling physically hungry; - Eating alone due to embarrassment/ embarrassment in eating; - Feeling disgusted with oneself and depressed or guilty after overeating; - Marked suffering from eating habits; - The lack of compensatory activities associated with it (inducing vomiting, using diuretics, or significantly increasing physical activity).	- Eating more or more than intended; - Persistent desire to eat or unsuccessful attempts to limit food consumption; - Devoting a lot of time to eating activities; - Neglecting social responsibilities and activities; - Eating food despite negative physical, mental, and social consequences; - Limiting or abandoning due to eating at important social events; professional or recreational activities; - The occurrence of withdrawal syndrome.	At least three of the following symptoms: - Skipping breakfast due to a lack of appetite at least four times a week; - A strong need to eat between an evening meal and falling asleep at night; - Difficulty falling asleep or waking up from sleep at least four nights a week; - The conviction that food is needed as a condition for starting or returning to sleep; - Frequent worsening of mood in the evening; - Significant suffering or deterioration of functioning; - Lack of criteria for psychological bulimia and binge eating syndrome.

**Table 4 nutrients-15-01641-t004:** The risk of weight gain is associated with the use of neuroleptics [27,28].

Second-generation neuroleptics			
High	Moderate	Low	Very low
Clozapine Olanzapine	Asenapine Paliperidone Quetapine Risperidone Sertindole	Amisulpiride Brexpiprazole Cariprazine Lurasidone	Aripiprazole Ziprasidone
First-generation neuroleptics			
	Perazine Promazine Zuclopenthixol	Haloperidol Perphenazine Sulpiride	

**Table 5 nutrients-15-01641-t005:** The risk of weight gain related to antidepressant use [29,30].

High	Moderate	Low
Tricyclic antidepressants, including Amitriptyline, doxepin, and clomipramine; Mirtazapine; Mianserin; Paroxetine.	SSRIs other than paroxetine during long-term treatment	Trazodone; Vortioxetine; Agomelatine; Bupropion; Duloxetine; Venlafaxine.

**Table 6 nutrients-15-01641-t006:** Recommended assessment of metabolic risk in patients with schizophrenia [61,64,65].

	Before Treatment	After 6 Weeks	After 3 Months	Every 12 Months
Family history	X			
Smoking, physical activity, and eating habits	X	X	X	X
BMI	X	X	X	X
Waist circumference	X	X	X	X
Blood pressure	X	X	X	X
Fasting glucose level	X	X	X	X
Fasting lipid profile	X		X	X
ECG	X			

**Table 7 nutrients-15-01641-t007:** Barriers to the treatment of obesity.

**Family doctor barriers:**
The lack of adequate knowledge about obesity and the ability to communicate with the patient; The view among some physicians that obesity is the result of a lack of discipline on the part of the patient; The doctor’s lack of conviction as to the sense of treatment and the possibility of obtaining positive effects of treatment; Too many duties and short time of medical consultation (10 min in Poland); Organizational factors such as the lack of optimization of work and the lack of involvement of other members of the GP team.
**Patient barriers:**
Sometimes, they do not trust their doctor; Patient’s attitude in accepting overweight and obesity and poor motivation for treatment; The lack of faith and belief in the possibility of obtaining positive effects of treatment; The lack of consistency and perseverance in treatment; Type of work and established habits in the family; Economic reasons—new drugs used to treat obesity are expensive, and so are healthy food products.
**Barriers on the system side:**
The lack of a dietitian, psychotherapist, and physiotherapist in the family doctor’s team; The lack of specialist obesity treatment centers to which a family doctor could refer a patient.

**Table 8 nutrients-15-01641-t008:** Organizational scheme—visits of patient with obesity.

**During the first visit with a new patient, anthropometric measurements should be performed, and if obesity is diagnosed, then:**
1. Refer the patient to tests to diagnose obesity complications; 2. Assess the patient’s readiness for treatment; 3. Implement the treatment; 4. Treatment monitoring—follow-up visits with brief advice at each visit; 5. At least once a year, perform an assessment of obesity complications.
**In relation to patients with obesity already under care:**
1. Once a year, according to the recommendation of the National Health Fund, body weight measuring in all patients (e.g., before vaccinations or visits for other reasons); 2. Weight measuring before each visit for patients with obesity complications, especially cardiovascular diseases; 3. Encouraging body weight measurements at home, asking about body weight during teleconsultations; 4. At least once a year in patients with obesity complications, control examinations assessing complications; 5. Short advice at every visit, if possible.

## Data Availability

Not applicable.

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
