# Peer review of "Obesity in Adults: Position Statement of Polish Association for the Study on Obesity, Polish Association of Endocrinology, Polish Association of Cardiodiabetology, Polish Psychiatric Association, Section of Metabolic and Bariatric Surgery of the Association of Polish Surgeons, and the College of Family Physicians in Poland"

_nutrients, 2023, doi:10.3390/nu15071641_

Round 1

Reviewer 1 Report

Proposed Position Statement of Polish Medical associations is a comprehensive description of obesity in adults, as a chronic relapsing disease. It is addressed specifically to definition , causes and diagnosis of obesity, management of obesity and its complications. There are also recommendations for general practitioners, regional authorities, and Ministry of Health. The article is extensive, perhaps with an abundant range of detail. Below are listed some suggested corrections to the text.

In table 1: last column 3rd and 4th rows: any should be substitutes by none, rows 5 and 6 substitute "last" by "least"

In the paragraph below table 1, there is missing explanation what are "mild or moderate" or "severe" complications, listed in table 1

Pls., check grammatically last sentence in paragraph 3.2.,

in text row 357 .. by one unit, row 505 explain pls. abbreviation "ORG"

English language correction by a native speaker would be beneficial.

Author Response

We are very grateful indeed for the efforts the Reviewers have taken to assess and improve our paper ‘Obesity in Adults: Position Statement of Polish Association for the Study on Obesity, Polish Association of Endocrinology, Polish Association of Cardiodiabetology, Polish Psychiatric Association, Section of Metabolic and Bariatric Surgery of the Association of Polish Surgeons and the College of Family Physicians in Poland’ (nutrients-2317375)  and we would like to thank you for yours thorough and detailed opinion and all suggestions and criticism.

The following corrections have been already attempted. All of them are marked red.

Reviewer 1

  1. In table 1: last column 3rd and 4th rows: any should be substitutes by none, rows 5 and 6 substitute "last" by "least"

Ad 1. Was corrected.

  1. In the paragraph below table 1, there is missing explanation what are "mild or moderate" or "severe" complications, listed in table 1

Ad 2. Was added “Determination of the severity of the disease according to the judgment of the clinician.”

  1. Pls., check grammatically last sentence in paragraph 3.2.

Ad 3.  Was corrected.

  1. in text row 357 .. by one unit,

Ad 4. Was added.

  1. row 505 explain pls. abbreviation "ORG"

Ad 5. Was explained.

  1. English language correction by a native speaker would be beneficial.

Ad 6. We corrected English language as best we could and we additionally requested a linguistic correction from MDPI.

Reviewer 2 Report

Dear Authors,

The article is very interesting and well designed. The initiative of providing such complex and useful Position Statement from a multidisciplinary perspective is excellent and it applies for medical and surgical specialists from all over the world, not only for family physicians, nurses, physiotherapists, registered dietitians, and psychologists due to its massive complexity. The domain of obesity in adults is extremely vast with a dramatic epidemiologic impact thus the importance of such guideline.

Here are my observations, suggestions or questions:

1.    It is not clear if the 11 authors are representing which of the 6 societies /associations. Did you have a working group or a panel of internal/external reviewers?

2.    Abstract – Line 32 – The research was focused around 6 main points (main topics). However, at Methods - line 77 – There are only 5 points.

3.    Introduction –  I suggest to cite WHO ICD-10 Classification

4.    Table 1 – How do you define pre-diabetes?

5.    Table 1 – Typo “at least” instead of “at last”

6.    Table 1- You should mention “mellitus” since there is “insipidus” type of diabetes, as well.

7.    Line 131 – Typo “and” instead of “nd”

8.    Table 2 and 3 should have a reference since they are not original research belonging to this paper. The same goes for Table 4 and 5.

9.    Table 3 – The symptoms should be listed in distinct lines.

10.  Lines 251-252 – “Diagnosis is not recommended in patients with iatrogenic Cushing's syndrome, especially those undergoing chronic glucocorticoid therapy.” Do you mean diagnostic tests as blood or urinary hormonal assays since clinical diagnostic and recognition is imperative in this situation?

11. Line 259 – I suggest to add “if confirmed endogenous hypercorti..”

12. Line 273 – I suggest to mention that the references are normal in adult general population since in children and teenagers are different and, also, outside pregnancy

13. Line 462 – I suggest to add “arterial” to “hypertension” at least in title of the subsection

14. Line 535 – I suggest to mention/add to “…hyperandrogenemia” in “women’

15. Line 540 –  FSH, LH and estradiol need to be assessed in specific days of the menstrual cycle

16. Section 3 – I suggest to mention sarcopenic obesity and osteoporosis/fragility fractures in patients with obesity, especially if type 2 diabetes mellitus is associated

Thank you

Author Response

We are very grateful indeed for the efforts the Reviewers have taken to assess and improve our paper ‘Obesity in Adults: Position Statement of Polish Association for the Study on Obesity, Polish Association of Endocrinology, Polish Association of Cardiodiabetology, Polish Psychiatric Association, Section of Metabolic and Bariatric Surgery of the Association of Polish Surgeons and the College of Family Physicians in Poland’ (nutrients-2317375)  and we would like to thank you for yours thorough and detailed opinion and all suggestions and criticism.

The following corrections have been already attempted. All of them are marked red.

  1. It is not clear if the 11 authors are representing which of the 6 societies /associations. Did you have a working group or a panel of internal/external reviewers?

Ad 1. As a leaders of this initiative Polish Associations Polish Association for the Study on Obesity delegated 3 members of the working group and Section of Metabolic and Bariatric Surgery of the Association of Polish Surgeons 4. Presidents of Polish Association of Endocrinology, Polish Association of Cardiodiabetology and Polish Psychiatric Association were the members of the working group and College of Family Physicians in Poland delegated the member of the obesitology section.

  1. Abstract – Line 32 – The research was focused around 6 main points (main topics). However, at Methods - line 77 – There are only 5 points.

Ad 2. Was corrected

  1. Introduction –I suggest to cite WHO ICD-10 Classification

Ad 3. The citation 2 was changed.

  1. Table 1 – How do you define pre-diabetes?

Ad 4. In accordance to definition impaired fasting glucose or/and impaired glucose tolerance. It was added.

  1. Table 1 – Typo “at least” instead of “at last”

Ad 5. Was corrected.

  1. Table 1- You should mention “mellitus” since there is “insipidus” type of diabetes, as well.

Ad 6. Was added.

  1. Line 131 – Typo “and” instead of “nd”

Ad 7. Was corrected.

  1. Table 2 and 3 should have a reference since they are not original research belonging to this paper. The same goes for Table 4 and 5.

Ad 8. Was added.

  1. Table 3 – The symptoms should be listed in distinct lines.

Ad 9. Was corrected.

  1. Lines 251-252 – “Diagnosis is not recommended in patients with iatrogenic Cushing's syndrome, especially those undergoing chronic glucocorticoid therapy.” Do you mean diagnostic tests as blood or urinary hormonal assays since clinical diagnostic and recognition is imperative in this situation?

Ad 10. Diagnosis was changed to Conducting laboratory hormonal tests

  1. Line 259 – I suggest to add “if confirmed endogenous hypercorti..”

Ad 11. Was added.

  1. Line 273 – I suggest to mention that the references are normal in adult general population since in children and teenagers are different and, also, outside pregnancy

Ad 12. Was added.

  1. Line 462 – I suggest to add “arterial” to “hypertension” at least in title of the subsection

Ad 13. Was added.

  1. Line 535 – I suggest to mention/add to “…hyperandrogenemia” in “women’

Ad 14. Was added.

  1. Line 540 – FSH, LH and estradiol need to be assessed in specific days of the menstrual cycle

Ad 15. Was added.

  1. Section 3 – I suggest to mention sarcopenic obesity and osteoporosis/fragility fractures in patients with obesity, especially if type 2 diabetes mellitus is associated

Ad 16. Thank you very much for this valuable comment, however, we have decided not to introduce these terms because we do not touch and define these terms, we decided not to introduce them so as not to cause consternation of readers.

The paper was corrected according to all Reviewers suggestions. We also tried our best to answer the Reviewer's doubts. We sent the new version of manuscript for further evaluation. We hope that the revision renders the manuscript acceptable for publication.